# Reinforcing Real-world Service Agents:
# Balancing Utility and Cost in Task-Oriented Dialogue

**Ning Gao** [* 1]  **Wei Zhang** [* 1]  **Yuqin Dai** [1]  **Ling Shi** [1]  **Ziyin Wang** [1]  **Yujie Wang** [1]  **Wei He** [1]  **Jinpeng Wang** [† 1]
**Chaozheng Wang** [⊠ 1]

## Abstract

The rapid evolution of Large Language Models (LLMs) has accelerated the transition from conversational chatbots to general agents. However, effectively balancing empathetic communication with budget-aware decision-making remains an open challenge. Since existing methods fail to capture these complex strategic trade-offs, we propose InteractCS-RL, a framework that reframes task-oriented dialogue as a multi-granularity reinforcement learning process. Specifically, we first establish a User-centric Interaction Framework to provide a high-fidelity training gym, enabling agents to dynamically explore diverse strategies with persona-driven users. Then, we introduce Cost-aware Multi-turn Policy Optimization (CMPO) with a hybrid advantage estimation strategy. By integrating generative process credits and employing a PID-Lagrangian cost controller, CMPO effectively guides the policy to explore Pareto boundary between user reward and global cost constraints. Extensive experiments on customized real business scenarios demonstrate that InteractCS-RL significantly outperform other baselines across three evaluation dimensions. Further evaluation on tool-agent-user interaction benchmarks verify InteractCS-RL robustness across diverse domains. The source code and data are available at https://github.com/NingGao-Ai/InteractCS-RL.

## 1. Introduction

With the rapid advancement of Large Language Models (LLMs), customer service systems have shifted the paradigm from simple intent classification to end-to-end generative agents. Building on this foundation, Task-Oriented Dialogue (TOD) methods have evolved, primarily focusing on helping users complete specific tasks such as booking restaurants, or reserving flights(Yao et al., 2024). Despite these advances, current methods typically involve simple information queries and form-filling conversations(Budzianowski et al., 2018), where the system only needs to follow fixed procedures to collect information, query databases, and return results (Bocklisch et al., 2024). However, real-world customer service scenarios are far more complex, filled with unexpected situations and emotional user interactions (Jun & Lee, 2025). This requires the agent to possess not only domain expertise but also refined verbal strategies and emotional regulation capabilities to alleviate user frustration. Crucially, these goals must be achieved under strict operational constraints, such as budget limits and efficiency targets, forming a multi-dimensional trade-off that existing general-purpose chatbots often struggle to balance.

Despite their conversational fluency, current TOD approaches often focus solely on problem resolution while neglecting cost optimization, leading to two critical limitations: (1) **Suboptimality of static data**: Existing methods rely on Supervised Fine-Tuning (SFT) over static dialogue corpora (Zhu et al., 2025; Ou et al., 2024; Li et al., 2025). This paradigm encourages models to imitate human behaviors, including cost-inefficient decisions such as premature concessions, without assessing their long-term optimality. As a result, these models are prone to error accumulation and struggle to generalize to dynamic environments that require strategic trade-offs. (2) **Lack of cost modeling mechanisms**: current frameworks predominantly utilize "success rate" as a monolithic metric. They fail to penalize "false successes", namely resolutions achieved through excessive resource expenditure, such as unnecessary compensation or protracted dialogue, thereby overlooking the economic constraints of real-world deployment.

To address these issues, we first reframe task-oriented dialogue as a *multi-granularity reinforcement learning process*. Instead of fixating on a single terminal objective, we argue that an ideal service agent should possess the intrinsic ca-

---

[*]Equal contribution    [†] Team leader. [1]Meituan, Beijing, China. Correspondence to: Chaozheng Wang <adf111178@gmail.com>.

*Proceedings of the 43rd International Conference on Machine Learning*, Seoul, South Korea. PMLR 306, 2026. Copyright 2026 by the author(s).

pacity to reconcile three simultaneous goals: maintaining service norms at every turn, resolving problems by session end, and controlling operational costs throughout.

Building upon this insight, we propose InteractCS-RL, a multi-granularity reinforcement learning framework for dynamic TOD.

Departing from traditional paradigms centered on static trajectory imitation, our framework establishes a closed-loop interactive evolution cycle, empowering the customer service agent to autonomously explore optimal strategies within a simulated business environment. Specifically, we first construct the **User-centric Interaction Framework**, which driven by realistic user profiles with intrinsic traits and extrinsic demands modeling to provide a high-fidelity interactively online training gym. At the algorithmic level, we propose **Cost-aware Multi-turn Policy Optimization** (CMPO), which leverages a hybrid advantage estimation strategy to provide multi-granular guidance: (1) **Session-level Outcome Utility** based on final task score, such as user satisfaction; (2) **Process Credit Assignment** using generative reward model to ensure each-turn conversational quality; (3) **Cost-aware Lagrange Penalty**, which transforms global constraints into dynamic cost signals. This design enables InteractCS-RL to reduce operational costs while maintaining high resolution rates, truly learning to solve problems in a "cost-conscious" manner.

We conduct comprehensive training and evaluation of InteractCS-RL on the representative real-world business scenarios called *FoodDeliveryService*. Experimental results demonstrate that our method significantly outperforms SFT, RL baselines (Shao et al., 2024; Xie et al., 2025) and SOTA closed-source models (OpenAI, 2025) across three evaluation dimensions. Furthermore, we perform cross-domain evaluation on public tool-agent-user benchmarks (Barres et al., 2025). Results show that InteractCS-RL maintains strong generalizability when handling multi-turn tool usage tasks across different domains. Finally, detailed ablation studies indicate the indispensability of both the profile-driven dynamic interaction environment and the cost-sensitive reward mechanism for enhancing agent decision-making capabilities in complex scenarios.

Our main contributions include:

- We reframe task-oriented dialogue as a multi-granularity reinforcement learning process, transitioning from traditional static trajectory imitation to a closed-loop interactive evolution cycle.
- We design the User-centric Interaction Framework driven with diverse persona profiles bank to provide real-world service scenarios.
- We propose Cost-aware Multi-turn Policy Optimization to ensure stable policy convergence and effectively inter-

nalize operational budgets into the agent decision-making process.
- Experiments show that InteractCS-RL significantly outperforms SOTA closed-source models and other baselines in both task resolution and cost-efficiency under the FDS scenario, and further demonstrate cross-domain generalizability on $\tau^2$-bench.

**Conflict of Interest Disclosure**   All authors are, or were at the time of this research, employed by or interning at Meituan, whose model, LongCat-Flash, was among those evaluated in this paper.

## 2. Related Works

**Task-oriented Dialogue** (TOD) systems are foundational to applications such as e-commerce, customer service, and automated sales(Deng et al., 2024; 2025). These systems must navigate complex interactions, ranging from fulfilling specific user requests to managing non-cooperative negotiations. Early research in TOD primarily leveraged sequence-to-sequence modeling and neural architectures(Vinyals & Le, 2015; Wen et al., 2015; Shang et al., 2015; Li et al., 2016a), often employing user simulators to augment training data(Li et al., 2016b; Lewis et al., 2017; Wei et al., 2018). However, the efficacy of these methods was largely constrained by the representative power of the underlying models. The advent of pre-trained and instruction-aligned LLMs has marked a paradigm shift(Naveed et al., 2023; Wang et al., 2024; Yao et al., 2023), as their sophisticated reasoning and linguistic capabilities are intrinsically well-suited for TOD. Current mainstream approaches typically focus on the construction of static datasets for supervised fine-tuning or offline reinforcement learning to adapt LLMs to specific scenarios (Li et al., 2025; Ou et al., 2024; Zhu et al., 2025; Bernard & Balog, 2023; Yu et al., 2023). Recent extensions have further integrated external tools (Peiyuan et al., 2024), RAG (Xu et al., 2024), multimodal inputs (Wang et al., 2025; Gong et al., 2025), and graph-guided evaluation (Shi et al., 2026) to broaden the boundaries of these agents. More recently, frameworks such as (Dai et al., 2026) and (Zhang et al., 2026) have advanced proactive TOD by integrating self-evolving user profiles and latent user concern modeling. Diverging from these proactive paradigms, InteractCS-RL focuses on passive TOD scenarios, establishing a dynamic online training environment driven by user persona role-playing.

**Rewards Utility in LLM Post-training.** Post-training for LLMs has evolved from simple preference alignment to the optimization of long-horizon reasoning. Early efforts focused on SFT to establish foundational capabilities (Radford et al., 2018; Mann et al., 2020). To improve generalization, Reinforcement Learning from Human Feedback (Christiano

et al., 2017; Ouyang et al., 2022) and its offline variants (Rafailov et al., 2023; Meng et al., 2024) established the standard for alignment. Furthermore, (Wan et al., 2026; Zhou et al., 2025; 2024) have been improved for multi-turn dialogues in RLHF. Recently, the Reinforcement Learning from Verifiable Rewards (RLVR) paradigm has gained prominence in objective domains like mathematics and coding. Concurrently, Group Relative Policy Optimization (GRPO) (Shao et al., 2024) and its derivatives (Yu et al., 2025; Zheng et al., 2025) have enhanced training stability and memory efficiency by eliminating the need for a centralized value network. However, these methods face two key challenges in complex TOD scenarios. To distribute sparse rewards, credit assignment (Sutton, 1984) is employed. Some approaches leverage step-level (Kazemnejad et al., 2024; Feng et al., 2025) or semantic-level (Guo et al., 2025) sampling reuse to improve advantage estimation, but they often fail to accurately assess response quality across multiple turns. To perform optimization under operational constraints, safe alignment (Ji et al., 2025) methods adopt offline optimization (Kim et al., 2025; Wachi et al., 2024) or Lagrange multiplier (Dai et al., 2023), they typically focus on the cost(Si et al., 2025) of individual responses rather than enforcing global constraints. To address these challenges in complex TOD scenarios, we introduce Process Credit Assignment and Cost-aware Lagrange Penalty, enabling more effective constraint-aware optimization in multi-turn dialogues.

## 3. Preliminaries

We formalize the interaction of task-oriented multi-turn dialogue as a turn-level Markov Decision Process $\mathcal{M} = \{\mathcal{S}, \mathcal{A}, P, R, \gamma\}$, where $\mathcal{S}$ denotes the state space, $\mathcal{A}$ denotes the action space, $P$ represents the transition dynamics, $R$ is the reward function and $\gamma$ is the discount factor. Specifically, the LLM serves as the agent $\pi_\theta$. The initial state $s_0$ is composed of the system instruction prompt $x_0$ and first response of user $u_0$, such that $s_0 = [x_0, u_0]$. As the interaction progresses, the state at time $t$, denoted as $s_t$, represents the cumulative dialogue trajectory: $s_t = [x_0, u_0, a_0, \ldots, u_{t-1}, a_{t-1}, u_t]$.

The action of agent at time $t$ is defined as a composite output $a_t = [z_t, y_t, d_t]$, where $z_t$ denotes reasoning process(Wei et al., 2022), $y_t$ represents the explicit response to the user, and $d_t$ signifies its underlying decision or discrete action. The state transition probability $\mathcal{P}(s_{t+1}|s_t, a_t)$ is primarily governed by user behavior; upon the execution of $a_t$, the environment generates the subsequent user response $u_{t+1}$. This transition $\mathcal{P}$ effectively captures the inherent stochasticity of real-world multi-turn dynamics. The reward function $r_t(s_t, a_t)$ reflects the utility gained during the interaction, and the objective of agent $J_R$ is to maximize the expected cumulative reward:

$$J_R(\pi_\theta) = \mathbb{E}_{\tau \sim \pi_\theta}\left[\sum_t \gamma^t r_t(s_t, a_t)\right]. \quad (1)$$

To simulate resource constraints or policy boundaries inherent in task-oriented scenarios, we further introduce a cost function $c_t(s_t, a_t)$, which quantifies the penalty incurred by the specific decision $d_t \sim \pi_\theta(a_t|s_t)$. Consequently, we model the TOD task as a Constrained Markov Decision Process (CMDP) (Altman, 2021). We emphasize the aggregate cost $J_C(\pi_\theta)$, where the optimization objective is to maximize task utility subject to the global cost threshold $\delta$:

$$\max_{\pi_\theta} \quad J_R(\pi_\theta), \quad \text{s.t.} \quad J_C(\pi_\theta) \leq \delta, \quad (2)$$

where $J_C(\pi_\theta) = \mathbb{E}_{\tau \sim \pi_\theta}\left[\sum_t \gamma^t c_t(s_t, a_t)\right]$ .This formulation compels the LLM agent to not only pursue task resolution but also to learn a fine-grained calibration of action costs within its policy distribution during interaction.

## 4. Method

We propose InteractCS-RL to address two critical gaps in the Task-Oriented Dialogue domain: 1) the absence of dynamic interactive environments featuring non-cooperative users and decision-making costs, and 2) the lack of online optimization methods to balance and cost in complex multi-turn dialogues. As illustrated in Figure 1, InteractCS-RL comprises two primary components: the **User-Centric Interaction Framework** and **Cost-aware Multi-turn Policy Optimization**. The former establishes a dynamic interaction environment by coupling intrinsic persona dimensions with extrinsic demands, while the latter employs a generative credit assignment mechanism and cost-aware lagrange penalty to enable the agent to balance service utility and operational costs. We detail these components and their implementation in the subsequent sections.

### 4.1. User-Centric Interaction Framework

To address the limitations of static datasets, we establish a dynamic, closed-loop interaction environment driven by realistic user personas and business scenario logic.

#### 4.1.1. PERSONA PROFILES BANK

To capture the psychological complexity of real-world users, we construct a standardized persona framework. We utilize LLMs to extract features from anonymized, high-quality business dialogue logs, distilling them into a bi-level profile:

**Intrinsic Traits** ($\mathcal{P}_{int}$)**:** Leveraging behavioral research (Cobb-Clark & Schurer, 2012), we model users across four stable dimensions:

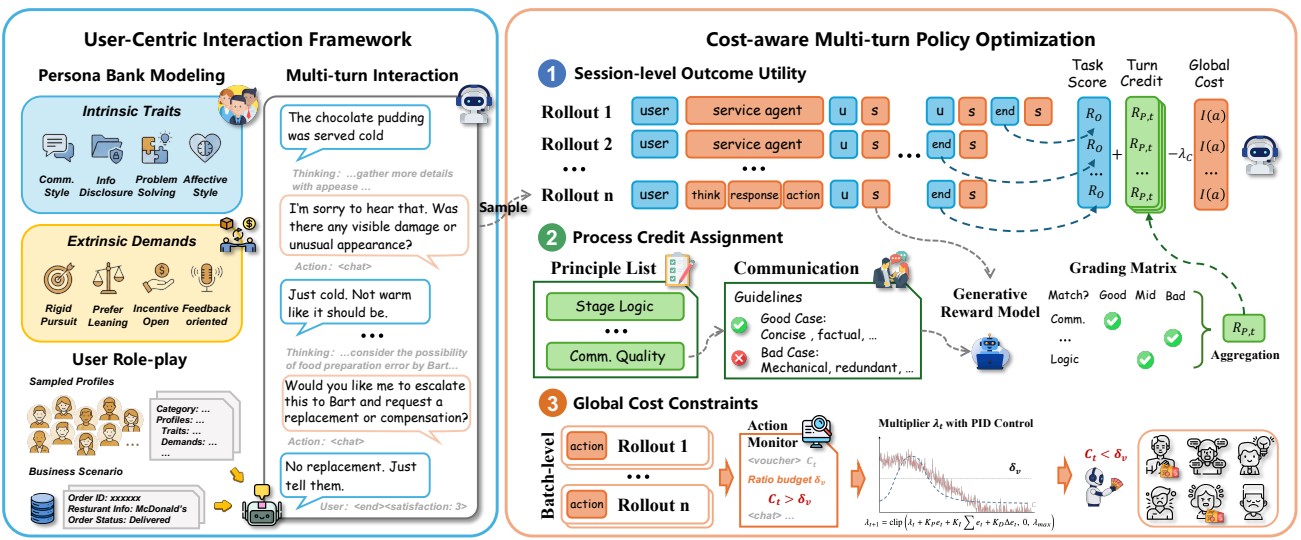

*Figure 1.* Illustration of our InteractCS-RL. (a) User-Centric Interaction Framework: Integrates persona bank modeling with dynamic user role-play to generate diverse interactive trajectories.(b) Cost-aware Multi-turn Policy Optimization: Synthesizes session-level outcomes, turn-level generative process credits, and PID-regulated global cost constraints into a hybrid advantage for stable policy optimization.

◇ *Communication Style:* Rhetorical features such as verbosity and interaction frequency.

◇ *Information Disclosure:* The user's initiative and completeness in providing privately key facts.

◇ *Problem-solving Style:* Strategic preferences in conflict, ranging from active collaboration to extreme not.

◇ *Personal Affective Style:* Psychological baseline and emotional stability of the specific user.

**Extrinsic Demands** ($\mathcal{D}$)**:** While intrinsic traits govern style, extrinsic demands capture the goal-oriented logic. We categorize user objectives into four distinct patterns: *Rigid Pursuit*, *Preference-learning*, *Incentive-open*, and *Feedback-oriented*.

By deeply coupling intrinsic traits and extrinsic demands, InteractCS-RL can automatically construct massive and diverse user behavior models, providing a diverse environment for agent optimization. More details please refer to App.E.1.

### 4.1.2. USER SIMULATION WITH ROLE-PLAY

We then employ LLMs $\pi_{user}$ as the user simulator. To ensure high fidelity, the simulator is conditioned on the constructed intrinsic features $\mathcal{P}_{int}$, extrinsic demands $\mathcal{D}$, and real-time business signals $S_{sys}$ including business scenario messages. By integrating these profiles directly as a structured prompt, the model dynamically generates linguistic responses and simulates psychological state transitions based on the conversation history $h_t = [u_0, y_0, ...u_{t-1}, y_{t-1}]$. The generation process is formalized as:

$$(u_{t+1}, m_{t+1}) = \pi_{user}(\cdot \mid h_t, \mathcal{P}_{int}, \mathcal{D}, S_{sys}), \quad (3)$$

where $u_t$ is the generated natural language response, and $m_t$ is metadata containing a satisfaction rating $\mathcal{S}$ and a dialogue termination signal $e$ as shown in the Fig.1.

### 4.1.3. MULTI-TURN DYNAMIC INTERACTION

The interactions in InteractCS-RL are modeled as a dynamic process of alternating agent decision-making and user feedback, designed to simulate the uncertainty present in real-world TOD scenarios. The agent receives the environmental state $s_t$ and generates composite $a_t$. The environment then parses this output and triggers a simulator response. The input-output relationship is defined as follows:

$$\begin{cases} \text{User Output: } (u_t, m_t) \sim \pi_{user}(h_t, \mathcal{P}_{int}, \mathcal{D}, S_{sys}), \\ \text{Agent Output: } a_t = [z_t, y_t, d_t] \sim \pi_\theta(s_t). \end{cases}$$
$$(4)$$

When the termination signal $e = 1$, the interaction terminates and the satisfaction score $\mathcal{S}$ from $\pi_{user}$ for task resolution is returned. This interaction framework not only provides the policy with the opportunity to explore the non-deterministic state space, but also supports the reinforcement learning process by collecting trajectory data with feedback.

## 4.2. Cost-aware Multi-turn Policy Optimization

Leveraging the interactive environment established in Sec.4.1, we formulate the task resolution process as a Constrained Markov Decision Process as illustrated in Sec.3.

Our goal is to train a customer service agent that maximizes task utility while strictly adhering to operational cost boundaries. To optimize this objective efficiently, we first adopt Group Relative Policy Optimization (GRPO) (Shao et al., 2024). Rather than relying on a learned value function, GRPO utilizes group-based rollouts to estimate the advantage. Departing from standard GRPO which relies solely on sparse outcome labels, our core contribution lies in the design of a **Hybrid Advantage Estimation strategy**. This mechanism unifies session-level outcomes, turn-level process guidance, and global cost constraints into a single learning signal. Formally, for the $i$-th trajectory $\tau_i$ sampled in a group, the advantage $\hat{A}_{i,t}$ at turn $t$ is computed as:

$$\hat{A}_{i,t} = \text{Norm}\left(\underbrace{\mathcal{R}_{O,i}}_{\text{Outcome}} + \underbrace{\mathcal{R}_{P,i,t}}_{\text{Process}} - \underbrace{\lambda \cdot \mathbb{I}(d_{i,t})}_{\text{Cost Penalty}}\right), \quad (5)$$

where Norm denotes normalizing operation on $R_{i,t}$ using the mean and standard deviation of all turn rewards, $\mathcal{R}_{O,i}$ represents the final user satisfaction, $\mathcal{R}_{P,i,t}$ denotes the turn-level process reward derived from principle adherence, and the cost term $\lambda \cdot \mathbb{I}(d_{i,t})$ applies dynamic penalties to high-cost actions (e.g, compensation). In the following sections, we detail the derivation and implementation of these three components.

### 4.2.1. SESSION-LEVEL OUTCOME UTILITY

The primary objective of a service agent is to resolve user issues effectively. We quantify this using the Outcome Reward ($\mathcal{R}_O$), which serves as the anchor for policy optimization. At the conclusion of each dialogue trajectory $\tau_i$, the user simulator $\pi_{user}$ acts as an evaluator, assigning a normalized satisfaction score based on the resolution status and the user's final emotional state. We directly utilize this score as the session-level reward: $\mathcal{R}_{O,i} = S_{i,final}$. This ensures that the agent's optimization direction remains consistently aligned with the ultimate business goal.

### 4.2.2. PROCESS CREDIT ASSIGNMENT

Reliance on sparse terminal signals often obscures the contribution of intermediate reasoning steps (Kazemnejad et al., 2024), making it difficult for agents to learn precise behavioral norms. To bridge this gap, we introduce a turn-level Process Reward ($\mathcal{R}_P$) mechanism inspired by Generative Reward Modeling (GenRM) (Liu et al., 2025).

We first translate domain expertise—such as dialogue logic, empathy requirements, and business compliance—into a set of scalable evaluation principles $P = \{p_1, p_2, \ldots, p_M\}$. We then employ an off-the-shelf LLM ($\pi_{\text{GenPRM}}$) as the auditor (detailed in Section F.4). For the $t$-th turn in trajectory $\tau_i$, the model evaluates the turn-level output $a_{i,t} \sim \pi_\theta(s_{i,t})$ against each principle $p_j$ with inference reasoning. To en-

sure the reliability of given process rewards, the evaluation score $S_{i,t,j}$ is constrained to a simple discrete set:

$$S_{i,t,j} = \pi_{\text{GenPRM}}(a_{i,t} \mid s_{i,t}, p_j) \in \{0, 0.5, 1\}, \quad (6)$$

where the discrete values correspond to violation, partial adherence, and full compliance. Finally, we aggregate these multi-dimensional assessments via a weighted summation to produce the turn-level process reward: $\mathcal{R}_{P,i,t} = \sum_{j=1}^{M} w_j \cdot S_{i,t,j}$. This mechanism provides fine-grained guidance, encouraging the agent to adhere to service specifications at every step of the interaction, rather than solely optimizing for the final outcome.

### 4.2.3. GLOBAL COST CONSTRAINTS

Here, we address the challenge of operational costs, such as excessive compensation, which must be controlled within a predefined budget. We solve this Constrained MDP by reformulating it into an unconstrained dual problem using the Lagrange multiplier method(Bertsekas, 2014; Achiam et al., 2017), a technique for finding local maxima and minima of a function over a constrained set. This allows us to transform the constrained primal problem defined in Eq.2 into its unconstrained Lagrangian dual form as follows:

$$\min_{\lambda \geq 0} \max_{\theta} \mathcal{L}(\theta, \lambda) = J_R(\pi_\theta) - \lambda(J_C(\pi_\theta) - \delta), \quad (7)$$

where $\lambda$ is the multiplier that dynamically penalize at the different training step.

Standard error-based updates of $\lambda$ often suffer from oscillation and overshooting. To ensure stable convergence between utility maximization and constraint satisfaction, we introduce a PID controller(Stooke et al., 2020). For the $k$-th step, we first compute the error term $e_k = J_{C,k}(\pi_\theta) - \delta$, where $J_{C,k}$ equals to the current average cost at $k$-th step, and then update the penalty coefficient $\lambda$ as follows:

$$\lambda_{k+1} = \text{clip}\left(\lambda_k + K_P e_k + K_I \sum e_k + K_D \Delta e_k, \; 0, \; \lambda_{max}\right). \quad (8)$$

By incorporating a proportional term to respond to instantaneous violations and an integral term to correct long-term bias, this mechanism effectively transforms global budget constraints into the stable penalty term used in Eq. 5, compelling the model to learn cost-aware planning.

Finally we derive the objective function below, combining with Eq.5 and Eq.8 to iteratively update the policy model:

$$\begin{aligned}
\mathcal{J}_{CMPO}(\theta) = & \mathbb{E}_{\mathcal{I} \sim Q, \{\tau_i\}_{i=1}^n \sim \pi_{\theta_{\text{old}}}} \left[\frac{1}{n} \sum_{i=1}^{n} \frac{1}{T_i} \sum_{t=1}^{T_i} \min\left(r_{i,t}(\theta)\hat{A}_{i,t}, \right.\right. \\
& \left.\left. \text{clip}(r_{i,t}(\theta), 1-\epsilon, 1+\epsilon)\hat{A}_{i,t}\right) - \beta \mathbb{D}_{KL}[\pi_\theta \| \pi_{\text{ref}}]\right].
\end{aligned} \quad (9)$$

*Table 1.* Comparison of model performance across different scenes on *FoodDeliverService*. Sat.: User satisfaction (Score from 1 to 5); FR: Dialogue Finish Rate(%); Comm.: Communication Quality (Score from 0 to 28); Logic: Logic Quality (Score from 0 to 12); V-Rate(%): Voucher Rate (Constrained with < 30%). Standard deviations are shown as superscripts.

| Method | Scene 1 (Hard) | | | | | Scene 2 (Easy) | | | | |
| | Task Score | | Dialogue Metric | | Cost | Task Score | | Dialogue Metric | | Cost |
| | Sat. ↑ | FR (%) ↑ | Comm. ↑ | Logic ↑ | V-Rate ↓ | Sat. ↑ | FR (%) ↑ | Comm. ↑ | Logic ↑ | V-Rate ↓ |
|---|---|---|---|---|---|---|---|---|---|---|
| *Large Models* | | | | | | | | | | |
| GPT-4.1 | $1.91^{\pm 0.19}$ | $83.8^{\pm 5.7}$ | $25.29^{\pm 0.48}$ | $10.59^{\pm 0.04}$ | $70.0^{\pm 3.3}$ | $2.12^{\pm 0.08}$ | $83.8^{\pm 4.5}$ | $25.67^{\pm 0.41}$ | $10.57^{\pm 0.04}$ | $70.4^{\pm 3.8}$ |
| DeepSeek-v3.2 | $1.96^{\pm 0.07}$ | $89.6^{\pm 1.9}$ | $\underline{26.67}^{\pm 0.19}$ | $\underline{10.68}^{\pm 0.12}$ | $76.7^{\pm 4.0}$ | $2.21^{\pm 0.20}$ | $87.9^{\pm 1.9}$ | $\underline{26.57}^{\pm 0.46}$ | $\underline{10.65}^{\pm 0.14}$ | $81.7^{\pm 1.4}$ |
| Qwen3-235B | $1.73^{\pm 0.07}$ | $62.5^{\pm 3.5}$ | $23.75^{\pm 0.27}$ | $7.82^{\pm 0.34}$ | $91.9^{\pm 4.4}$ | $1.93^{\pm 0.03}$ | $60.6^{\pm 2.7}$ | $23.80^{\pm 0.42}$ | $7.29^{\pm 0.39}$ | $94.2^{\pm 1.4}$ |
| LongCat-Flash | $2.18^{\pm 0.15}$ | $91.7^{\pm 3.1}$ | $25.12^{\pm 0.47}$ | $10.00^{\pm 0.09}$ | $93.3^{\pm 0.7}$ | $2.32^{\pm 0.14}$ | $90.4^{\pm 1.4}$ | $25.45^{\pm 0.26}$ | $9.84^{\pm 0.27}$ | $93.3^{\pm 0.7}$ |
| *Foundation Models* | | | | | | | | | | |
| Qwen-2.5-7B | $2.13^{\pm 0.15}$ | $53.3^{\pm 5.1}$ | $22.50^{\pm 0.49}$ | $6.87^{\pm 0.15}$ | $77.5^{\pm 3.8}$ | $2.27^{\pm 0.16}$ | $55.5^{\pm 5.0}$ | $23.20^{\pm 0.66}$ | $6.63^{\pm 0.19}$ | $75.4^{\pm 4.4}$ |
| Qwen-2.5-14B | $1.88^{\pm 0.00}$ | $76.2^{\pm 4.8}$ | $22.89^{\pm 0.68}$ | $8.92^{\pm 0.05}$ | $37.9^{\pm 4.7}$ | $2.09^{\pm 0.12}$ | $80.0^{\pm 3.3}$ | $23.62^{\pm 0.16}$ | $8.90^{\pm 0.13}$ | $29.6^{\pm 1.9}$ |
| *Static Train* | | | | | | | | | | |
| Qwen-2.5-7B-SFT | $2.00^{\pm 0.10}$ | $94.2^{\pm 3.1}$ | $24.15^{\pm 0.29}$ | $9.64^{\pm 0.02}$ | $41.7^{\pm 0.7}$ | $2.16^{\pm 0.22}$ | $94.2^{\pm 1.4}$ | $24.09^{\pm 0.51}$ | $9.42^{\pm 0.06}$ | $42.9^{\pm 4.4}$ |
| Qwen-2.5-14B-SFT | $2.15^{\pm 0.09}$ | $99.6^{\pm 0.7}$ | $24.63^{\pm 0.50}$ | $10.11^{\pm 0.24}$ | $45.0^{\pm 4.5}$ | $2.13^{\pm 0.06}$ | $98.8^{\pm 2.2}$ | $24.36^{\pm 0.55}$ | $9.95^{\pm 0.17}$ | $44.6^{\pm 2.9}$ |
| *InteractCS-RL (Ours)* | | | | | | | | | | |
| Qwen-2.5-7B-RL | $\underline{2.74}^{\pm 0.10}$ | $\underline{100.0}^{\pm 0.0}$ | $26.12^{\pm 0.25}$ | $10.07^{\pm 0.25}$ | $30.8^{\pm 1.4}$ | $2.82^{\pm 0.02}$ | $100.0^{\pm 0.0}$ | $26.10^{\pm 0.18}$ | $10.27^{\pm 0.25}$ | $\underline{34.6}^{\pm 1.9}$ |
| Qwen-2.5-14B-RL | $\mathbf{3.05}^{\pm 0.04}$ | $\mathbf{100.0}^{\pm 0.0}$ | $\mathbf{27.43}^{\pm 0.28}$ | $\mathbf{11.34}^{\pm 0.21}$ | $27.5^{\pm 2.5}$ | $\mathbf{3.21}^{\pm 0.07}$ | $\mathbf{100.0}^{\pm 0.0}$ | $\mathbf{27.45}^{\pm 0.29}$ | $\mathbf{11.34}^{\pm 0.46}$ | $\mathbf{28.7}^{\pm 3.8}$ |

# 5. Experimental Setup

## 5.1. Evaluation Objectives

To evaluate the proposed framework, we design our experiments to answer the following three questions:

$\mathcal{H}1$ - Can our method achieve higher dialogue quality in real-world scenarios while maintaining acceptable cost levels?

$\mathcal{H}2$ - Is the proposed CMPO in InteractCS-RL effective?

$\mathcal{H}3$ - Can the model trained under the proposed setting generalize to other TOD dialogue scenarios?

## 5.2. Benchmarks and Evaluation Metrics

**Food Delivery Service (FDS).** We construct a high-fidelity food delivery after-sales dispute scenario as the primary evaluation benchmark. This scenario covers core dispute types, including delivery delays, damaged food, and missing or incorrect orders. The agent's objective is to resolve user dissatisfaction through multi-round negotiation while adhering to business compliance constraints. The primary cost incurred by the agent arises from voucher compensation actions. For the FDS scenario, we build a user profile library from pre-collected real-world business data and use it to train the SFT baseline. In addition, user profiles are clustered into five levels based on their cooperation degree, and two difficulty settings are designed by adjusting the proportion of profiles at different levels to simulate more diverse user populations. More details please refer to Appendix A.1.

**Evaluation Metrics.** Agent performance is evaluated along three dimensions. *Task score* includes the average user sat-isfaction and the completion rate of formatted dialogues. *Dialogue quality* measures the logical consistency and appropriateness of the agent expressions. *Voucher Rate* serves as the constraint indicator, capturing the frequency of issuing coupons during dialogue trajectories. All evaluations are conducted with three random tests. Part of metrics require the use of llm-as-a-judge to generate base scores, and we utilize the Deepseek-v3.2 model during the process. Additional details are provided in the Appendix A.1.

Additionally, we employ $\tau^2$-bench (Barres et al., 2025) to evaluate the generalizability of our InteractCS-RL across diverse TOD domains, including Retail, Telecom, and Airline. We assess the agent's performance using Pass@1, Communicate Rate (Comm. Rate), DB Rate, and Action Reward. The detailed experimental setup and metric definitions for this benchmark are provided in Appendix A.2.

## 5.3. Implementation Details

All components are implemented using the Verl distributed framework. The customer service agent is based on Qwen2.5-7/14B-Instruct, while the user role-playing model and reward generation model use Qwen2.5-32B-Instruct. The learning rate is set to $1 \times 10^{-6}$ with cosine annealing, a warmup ratio of 0.1, and an annealing ratio of 0.2. The sampling group size is $G = 4$, and the batch size is 128. Dialogues terminate when the user is satisfied, explicitly refuses, or reaches the maximum number of rounds $T_{\max} = 15$. Experiments are conducted on 8 NVIDIA A100 GPUs for service agent training and 2 NVIDIA H20 GPUs for other model inference. Detailed parameter settings please refer to Appendix E.4.

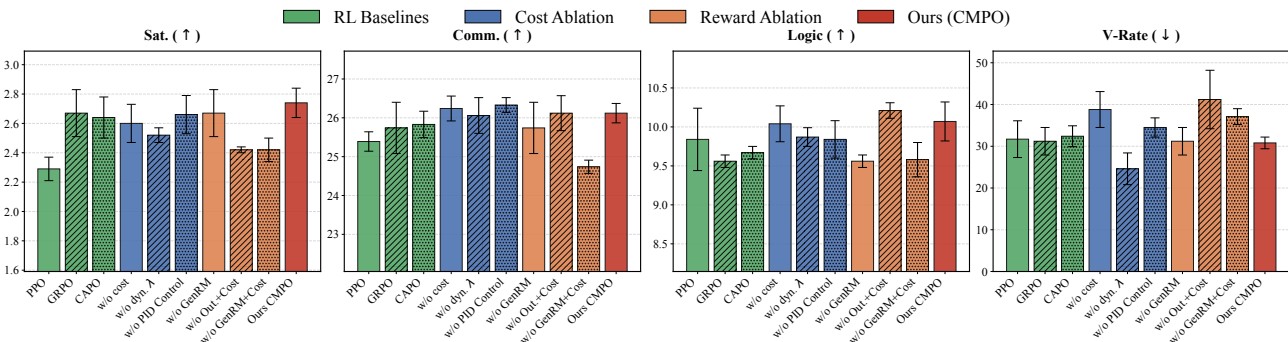

*Figure 2.* Comprehensive ablation studies validate the effectiveness of InteractCS-RL across three key dimensions: efficacy of CMPO, PID-controlled cost mechanisms, and hybrid reward formulations. For detailed numerical results corresponding to this figure, please refer to Appendix B.3.

## 5.4. Baselines

**Large models.** We evaluate several state-of-the-art closed-source and open-source models, including GPT-4.1(OpenAI, 2025), Deepseek-v3(Liu et al., 2024), LongCat-Flash(Team et al., 2025), and Qwen3-235B(Yang et al., 2025). **Base models.** Qwen2.5-Instruct (7B and 14B)(Yang et al., 2024) are evaluated with carefully designed business prompts. **SFT and RL models.** SFT is performed on approximately 2k high-quality samples, obtained through filtering and enhancement of pre-collected business data. For RL, we compare CMPO with PPO(Schulman et al., 2017), GRPO (Shao et al., 2024), and CAPO (Xie et al., 2025)

## 6. Experiment Results

### 6.1. Main Results

Table 1 presents the comparative experimental results of FDS scenarios with varying difficulty levels. We observe the following advances of our InteractCS-RL to answer $\mathcal{H}1$:

(1) **Significantly Improved Task Effectiveness:** In the Hard scenario, the 14B model trained with InteractCS-RL can improve user satisfaction to 3.05 points, surpassing closed-source models including LongCat-Flash by nearly 40%; it also achieves a 100.0% dialogue completion rate (FR), outperforming top-tier closed-source models GPT-4.1 (83.8%) and DeepSeek-v3.2 (89.6%).

(2) **High-Quality Dialogue Process:** InteractCS-RL achieves the highest scores in both logical quality (11.34) and communication quality (27.43), representing a 10% improvement over the SFT model in each area. This indicates that the round-based principle introduced by GenRM effectively constrains the agent's cognitive coherence during long-term interactions.

(3) **Precise Cost Awareness:** Under the preset constraint of Voucher Rate, all large models exhibited severe budget

overruns. Supervised fine-tuning models performed relatively better but still exceeded the threshold. Our method successfully controlled the payout ratio around the set threshold of 30%.

(4) **Universal Improvement Across Scale and Scenarios:** Through our InteractCS-RL, both scale models showed consistent performance improvements and stably adapted to scenarios of varying difficulty.

Overall, these results underscore the superiority of InteractCS-RL in internalizing complex service norms and cost boundaries, effectively breaking the performance ceiling of static imitation to achieve a better balance between service utility and operational economy. Furthermore, we supplement comparisons and discussions of some other methods in B.1.

### 6.2. Studies and Ablation on CMPO

In this section, we conduct a comprehensive analysis to validate the effectiveness of the proposed CMPO and its sub-components. We examine the impact of different reinforcement learning baselines, the necessity of the PID-controlled constraint mechanism, and the contribution of each reward component to answer $\mathcal{H}2$.

**Efficacy of CMPO.** We first compare InteractCS-RL against established RL baselines, including token-level PPO, session-level GRPO, and turn-level CAPO. Comparing CMPO with green lines, we can observe distinct trade-offs exist across different granularities. Token-level PPO achieves strong Logical Consistency (9.84) by optimizing immediate syntax probabilities but suffers significantly in goal completion (Sat. 2.29), indicating that dense token supervision struggles to capture long-horizon task utility. Conversely, standard GRPO improves Satisfaction (2.67) but exhibits degradation in process logic (9.56) due to the sparsity of outcome-only feedback. Our method, by inte-

*Table 2.* Performance of our InteractCS-RL in $\tau^2$-Bench.

| Methods | Retail | | | | Airline | | | | Telecom |
| --- | --- | --- | --- | --- | --- | --- | --- | --- | --- |
| | Pass@1 | Comm. Rate | DB Rate | Action Reward | Pass@1 | Comm. Rate | DB Rate | Action Reward | Pass@1 |
| Qwen2.5-7B | 14.4% | 61.4% | 16.6% | 152 | 14.0% | 76.0% | 18.0% | 60 | 8.8% |
| Qwen2.5-7B-SFT | 15.8% | 65.8% | 18.4% | 169 | 18.0% | 78.0% | 22.0% | 61 | 10.5% |
| InteractCS-RL | **21.1%** | **67.5%** | **23.7%** | **233** | **24.0%** | **82.0%** | **28.0%** | **65** | **14.9%** |
| Qwen2.5-14B | 44.7% | 80.7% | 45.6% | 337 | 16.0% | 88.0% | 20.0% | 75 | 17.5% |
| Qwen2.5-14B-SFT | 43.9% | 81.6% | 45.2% | 325 | 20.0% | 88.0% | 26.0% | 74 | 20.2% |
| InteractCS-RL (Ours) | **47.4%** | **85.1%** | **49.1%** | **355** | **28.0%** | **92.0%** | **32.0%** | **75** | **24.6%** |

grating turn-level principle rewards with cost-sensitive outcomes, achieves a Pareto optimal state. It not only secures the highest User Satisfaction (2.74) and Communication Quality (26.12) but also enforces the strictly defined cost adherence (V-Rate 30.8%). This demonstrates that CMPO successfully bridges the gap between myopic generation and sparse objective optimization.

**Impact of Cost Constraint Mechanisms.** A core contribution of our work is the PID-Lagrangian mechanism for global cost control. We analyze four settings in the blue part in Fig. 2 to understand the behavior of different constraint strategies, further visualization results are provided in the Appendix H:

*w/o Cost ($\lambda = 0$):* Without any penalty, the agent initially attempts to boost user satisfaction by finding a "shortcut" of overly distributing compensation. This results in a Voucher Rate of 38.8%, significantly violating the operational threshold ($< 30\%$). Morever, relying prematurely on this lazy strategy prevents the model from exploring advanced communication skills needed for complex users, ultimately resulting in a suboptimal overall Satisfaction (2.60).

*w/o PID Control:* When removing the PID controller and relying on a basic relative error update, the agent struggles to stabilize. The Voucher Rate oscillates and settles at 34.5%, failing to strictly satisfy the constraint.

*w/o Dynamic $\lambda$ (Fixed $\lambda$):* We also test a static penalty coefficient. Results show that a fixed penalty lacks the flexibility to adapt to training dynamics. It tends to over-suppress the agent's actions (V-Rate drops to 24.6%), which severely harms the user experience, resulting in the lowest Satisfaction (2.52) among the groups.

*Ours (CMPO):* By incorporating proportional and integral terms, our method dynamically adjusts $\lambda$ to correct both instantaneous errors and long-term bias. This allows the model to converge precisely near the constraint boundary (V-Rate 30.8%) while maximizing utility, ultimately yielding the highest Satisfaction (2.74).

**Ablation of Hybrid Reward Components.** Finally, we investigate the contribution of different reward signals in the orange part in Fig. 2. Removing the Outcome Utility and

Cost constraints (*w/o Out.+Cost*) results in a model that excels in Logical Consistency (10.21) but fails to resolve actual user problems (Sat. 2.42) or save costs (V-Rate 41.2%), as it only optimizes for conversational norms. Conversely, removing the Generative Reward Model (*w/o GenRM*) causes a drop in Logical Consistency to 9.56, as the model loses fine-grained guidance on empathy and procedure. The full CMPO integrates all components, demonstrating that high-quality service requires the simultaneous optimization of outcome utility, process logic, and cost constraints. Our approach achieves the best satisfaction under cost-aware conditions, validating the necessity of reward formulation.

### 6.3. Generalizability

In this section, we evaluate the generalizability of our InteractCS-RL in $\tau^2$-bench to answer $\mathcal{H}3$, with the results shown in Table 2.

As observed, the dual-control environment poses a significant challenge; standard models struggle to coordinate reasoning and action, with the Qwen2.5-7B Instruct model achieving less than 15% Pass@1 in the Retail and Airline domains. Furthermore, direct SFT demonstrates limited generalizability. In some cases, SFT even leads to performance degradation (e.g., Qwen2.5-14B-SFT drops from 44.7% to 43.9% in Retail), suggesting that static imitation learning fails to capture the underlying problem-solving logic required for unseen domains.

In contrast, InteractCS-RL consistently outperforms baselines across all datasets and model sizes. On average across the three domains, our method improves the average Pass@1 rate by 5.6% compared to the SFT baseline on the 14B model. Crucially, the simultaneous improvements in **DB Rate** and **Action Reward** indicate that our approach significantly enhances the model's ability to execute precise tool calls and adhere to complex domain constraints, even without specific training data for these domains. This suggests that our cost-aware reinforcement learning framework fosters robust, transferable reasoning capabilities rather than mere pattern matching.

Additionally, the consistent gains in **Communicate Rate** (e.g., +6.1% in Retail for 7B) imply that our training method-

ology encourages the agent to be more informative and proactive. This demonstrates that InteractCS-RL not only solves the technical aspects of the task but also elevates the conversational standard, making the agent more effective at delivering critical information to users in a service context.

### 6.4. Case Study

In Appendix C, we present case studies comparing InteractCS-RL with the SFT baseline to highlight its superior decision-making. In FDS scenarios (C.1), our method demonstrates refined adaptability to diverse user personas—successfully balancing empathetic appeasement with strict SOP and cost adherence—whereas SFT often falls into repetitive, ineffective dialogue loops. Furthermore, cross-domain evaluations on $\tau^2$-bench (C.2) show that InteractCS-RL better understands complex user intents and executes proactive guidance, overcoming the semantic gaps and mechanical stagnation prevalent in static SFT models.

## 7. Conclusion

In this paper, we proposed InteractCS-RL to address the critical tension between empathetic communication and operational cost in task-oriented dialogue. To reframe TOD as a multi-granularity reinforcement learning process, InteractCS-RL integrates a User-centric Interaction Framework for high-fidelity strategy exploration with CMPO, which employs a PID-Lagrangian controller to internalize global costs with turn-level generative process credit. Extensive experiments demonstrate that InteractCS-RL significantly outperforms state-of-the-art models, achieving better results between task score under budget constraints while maintaining robust performance across diverse domains.

## Impact Statement

This paper presents work whose goal is to advance the field of machine learning. There are many potential societal consequences of our work, none of which we feel must be specifically highlighted here.

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

# A. Benchmarks Description

## A.1. FoodDeliverService

### A.1.1. BENCHMARK DESCRIPTION

FoodDeliverService is a specialized evaluation environment simulating complex customer service negotiations within the food delivery domain. Unlike standard task-oriented dialogues where user intent is static, this benchmark models realistic, friction-heavy interactions where the agent must manage user dissatisfaction, negotiate compensation, and business Standard Operating Procedures (SOPs). The framework is formalized as a dynamic multi-turn interaction environment where the User Simulator is parameterized by a tuple $\mathcal{U} = \{\mathcal{P}_{int}, \mathcal{D}\}$, representing intrinsic behavioral traits and extrinsic demands. The agent's objective is to resolve complaints (e.g., cold food, missing items) while balancing user satisfaction against operational costs. You can find source data about the FoodDeliverService bench in this repo https://github.com/NingGao-Ai/InteractCS-RL.. Detailed specifications of the environment are provided in the Appendix E.

### A.1.2. EVALUATION METRICS

To comprehensively evaluate the agent's performance, we employ a three-tiered metric system comprising Task Scores, Dialogue Quality Metrics, and Cost Control, as detailed below:

- **Task Score:** This category evaluates the direct outcome of the interaction from the user's perspective and the agent's adherence to system constraints.

  - **User Satisfaction (Sat.):** A score ranging from 1 to 5, derived directly from the user simulator's feedback signal at the end of the episode. It reflects the user's emotional state and perceived resolution quality.
  - **Dialogue Finish Rate (FR):** A binary success metric (averaged as a percentage) that validates the agent's strict adherence to the required output format and pre-set rules. A session is considered "Finished" only if the agent correctly utilizes the XML-based thinking/action tags (e.g., `<action>voucher</action>`) and complies with hard constraints, such as issuing a voucher at most once per session.

- **Dialogue Metric:** We utilize a fine-grained "LLM-as-a-Judge" pipeline to evaluate the procedural quality of the conversation. The specific evaluation prompts used for this pipeline are detailed in the Appendix F.2.

  - **Communication Quality (Comm.):** A cumulative score (0–28) assessing the agent's linguistic and interpersonal performance. It aggregates scores across seven dimensions: Identity Neutrality, Dialogue Quality (novelty/focus), Language Adaptability, Content Quality, Communication Effectiveness (conciseness/sincerity), Natural Fluency, and Context Adaptability.
  - **Logic Quality (Logic):** A cumulative score (0–12) assessing the agent's reasoning and business logic. It evaluates three critical areas: User Profile Recognition (identifying emotions/intent), Business Rule Capability (SOP compliance, authenticity, fairness, consistency), and Out-of-Distribution (OOD) Issue Recognition.

- **Cost (Operational Efficiency):**

  - **Voucher Rate (V-Rate):** This metric quantifies the operational cost by measuring the percentage of sessions where the agent issues a financial compensation (voucher). To ensure sustainable service operations, agents are penalized if the V-Rate exceeds a threshold (e.g., $< 30\%$), encouraging negotiation over immediate monetary concession.

### A.1.3. IMPLEMENTATION DETAILS

The experimental setup for FoodDeliverService enforces a strict structured output format to facilitate automated parsing and reasoning evaluation. Agents are required to output a chain-of-thought process wrapped in `<think>` tags, followed by the response in `<response>` tags, and a final executable decision in `<action>` tags (values: *chat* or *voucher*).

To ensure consistent and objective evaluation of the Dialogue Metrics, we employ a specialized evaluator model prompted with the detailed scoring taxonomies described above. This evaluator analyzes the entire dialogue history to assign the Communication and Logic scores, ensuring that agents are rewarded not just for the final outcome, but for the empathy, logic, and safety of their conversational trajectory.

For the experimental setup, we utilize DeepSeek-V3.2 as the user simulator to ensure high-fidelity and diverse interactions, with assistant models deployed on two NVIDIA H20 GPUs. The test sample consisted of 80 users, and three random tests were conducted.

## A.2. $\tau^2$-Bench

### A.2.1. BENCHMARK DESCRIPTION

$\tau^2$-bench is a comprehensive evaluation framework designed to assess conversational agents in a dual-control environment, formalized as a Decentralized Partially Observable Markov Decision Process (Dec-POMDP). Distinct from traditional benchmarks where the user acts as a passive information provider, $\tau^2$-bench simulates realistic scenarios where both the agent and the user possess agency to employ tools and modify the shared world state. The benchmark specifically targets complex service domains, including Telecom, Retail, and Airline, requiring the agent not only to reason effectively about domain constraints but also to coordinate with and guide the user through necessary actions to resolve issues.

### A.2.2. EVALUATION METRICS

We employ a multi-dimensional metric system to evaluate the agent's performance:

- **Pass@1:** This serves as the primary indicator of task success, measuring the model's ability to fulfill user requirements. For the Airline and Retail domains, Pass@1 is strictly defined by the satisfaction of two rule-based sub-metrics:
  - **Communicate Rate (Comm. Rate):** This metric validates whether the agent has successfully conveyed all essential information required by the user (e.g., explaining policy details or confirming flight times).
  - **DB Rate:** This evaluates the integrity of the post-interaction database. It checks if the final state of the database (e.g., order status updated to "refunded") strictly matches the expected ground truth derived from the user's intent.

- **Action Reward:** Beyond the final outcome, this metric assesses the procedural quality of the dialogue. For each task, a sequence of expected actions is predefined (including both mandatory and optimal steps). The action reward quantifies the proportion of these expected actions successfully executed by the model, reflecting its adherence to standard operating procedures.

### A.2.3. IMPLEMENTATION DETAILS

For the experimental setup on this benchmark, we deploy the assistant models for inference using two NVIDIA H20 GPUs. To ensure high-fidelity and diverse user interactions during the evaluation, we utilize DeepSeek-V3.2 as the user simulator.

## B. Additional Experimental Results and Discussions

### B.1. Comparisons with Additional RL Baselines

To further evaluate the effectiveness of our proposed framework, we conduct additional comparisons with other Reinforcement Learning paradigms adapted for the Task-Oriented Dialogue domain. Specifically, we introduce two specialized baselines: a KRLS-style offline RL approach(Yu et al., 2023) and a curiosity-driven exploration approach(Wan et al., 2026). The experimental results, based on the Qwen2.5-7B backbone, are summarized in Table 3.

*Table 3.* Performance comparison of InteractCS-RL against additional RL baselines on Qwen2.5-7B.

| Model | Sat. | Comm. | Logic | FR (%) | V-Rate (%) |
|---|---|---|---|---|---|
| SFT | 2.00 | 24.15 | 9.64 | 94.2 | 41.7 |
| KRLS-style Offline RL | 2.46 | 25.93 | 9.92 | 100.0 | 44.5 |
| InteractCS-RL w/ CU Reward | 2.55 | 25.43 | 9.87 | 100.0 | 32.4 |
| **InteractCS-RL (Ours)** | **2.74** | **26.12** | **10.07** | **100.0** | **30.8** |

**Comparison with KRLS-style Offline RL.** The KRLS paradigm typically decomposes multi-turn conversations into single turns based on static answer sequences and applies offline RL. To adapt this to our scenario, we extracted static historical dialogues from the SFT dataset, split them in a KRLS-style manner, and required the model to generate single-turn

responses based on the static history. We applied a turn-level reward emphasizing generation quality and key entity matching, notably without any multi-turn simulation or global cost constraints. As shown in Table 3, while this offline RL approach outperforms the standard SFT model, it falls short of InteractCS-RL on all metrics except the completion rate (FR). More importantly, its coupon distribution rate (V-Rate) reaches 44.5%, significantly exceeding acceptable operational thresholds. This validates our core motivation: an end-to-end training paradigm within a dynamic, multi-turn environment is essential for effectively balancing user utility and service costs.

**Comparison with Curiosity-Driven Exploration.** We also implemented a baseline based on curiosity-driven exploration, denoted as InteractCS-RL w/ CU Reward. In this setup, we required the LLM to predict the user's profile category after each dialogue turn based on the history, converting the prediction error into an intrinsic curiosity reward. Cost constraints from our CMPO component were retained for a fair comparison. However, empirical results indicate that in real-world customer service scenarios, the agent tends to repeatedly ask for user details and artificially prolong the dialogue to harvest these curiosity rewards. This behavior directly leads to a decrease in user satisfaction (Sat. drops to 2.55). This demonstrates that while curiosity rewards are effective for open-ended exploration, they are misaligned with efficiency-oriented and cost-sensitive dialogue scenarios.

**Discussion on Other Multi-turn RL Paradigms.** Beyond the implemented baselines, it is worth discussing other RL paradigms for multi-turn planning. Early works, such as Deep Dyna-Q, address constrained dialogue planning but primarily rely on predefined dialogue action spaces and policies. In contrast, our approach tackles the problem in a more open-ended generation setting. Moreover, recent LLM-based RL agents like SWEET-RL(Zhou et al., 2025) and ArCHer(Zhou et al., 2024) represent distinct and valuable paradigms but are difficult to directly adapt to our specific setting. SWEET-RL requires comprehensive "training-time information" to train a highly accurate Critic, which is feasible in objective tasks like programming but inapplicable in negotiation dialogues where user psychological states are subjective and unobservable. ArCHer utilizes a hierarchical, off-policy architecture requiring multiple Q-Critics and V-Critics (including target networks), introducing prohibitive computational overhead at the 7B parameter scale. In summary, compared to these paradigms, InteractCS-RL focuses on a constrained balance learning framework. By synthesizing turn-level generative process credits and session-level outcomes, it achieves a mixed advantage that ensures stable policy optimization for real-world service agents.

## B.2. Human Evaluation on Simulator and Judge Quality

To ensure the fidelity of our proposed dynamic environment and the reliability of our automated metrics, we recruited six domain experts from a professional customer service team to conduct a comprehensive two-phase human evaluation.

**Study 1: User Simulator Fidelity.** First, we evaluated whether our simulator effectively mimics real user behavior. Each expert interacted with the user simulator as a service agent across 35 distinct dialogue sessions, covering diverse user profiles (yielding over 200 samples in total). They rated the simulator on a 5-point Likert scale across three dimensions: *Humanness*, *Decision Rationality*, and *Emotion*. As shown in Table 4(a), the simulator achieved exceptionally high scores (averaging 4.64 across all criteria), confirming its capability to faithfully replicate complex real-world customer behaviors.

*Table 4.* Human evaluation results. (a) Domain experts' ratings of the user simulator fidelity. (b) The highest human scores achieved by our CMPO agent and the Spearman's rank correlation ($\rho$) between human ratings and the LLM judge.

**(a) Simulator Fidelity (1-5)**

| Metric | Avg. Score |
| --- | --- |
| Humanness | 4.67 |
| Decision Rationality | 4.52 |
| Emotion | 4.72 |

**(b) Trajectory Scoring & Correlation**

| Dimension | CMPO | Corr. ($\rho$) |
| --- | --- | --- |
| Communication | 4.78 | 0.87 |
| Logic | 4.36 | 0.82 |

**Study 2: Blind Trajectory Scoring and Correlation.** Second, we assessed the generation quality of the agents and validated our LLM-as-a-judge methodology. We randomly sampled 40 test trajectories generated by the SFT baseline, GPT-4.1 (w/ SOP), and Qwen-2.5-14B (w/ CMPO). All trajectories were anonymized and fully shuffled. The experts independently scored each trajectory on *Communication* (Comm.) and *Logic* using a 5-point Likert scale. Consistent with our automated evaluations, Qwen-2.5-14B with CMPO achieved the highest human ratings across both dimensions (Table 4(b)). Furthermore, we computed the Spearman's rank correlation coefficient between these human scores and the automated scores provided by LLM-as-a-judge with DeepSeek-V3.2,. The strong correlations ($\rho = 0.87$ for Comm. and

0.82 for Logic) demonstrate a high degree of alignment with human preferences, firmly validating the reliability of our automated evaluation pipeline.

### B.3. Ablation Details

Specifically, the subsequent tables sequentially report the detailed metrics perfectly mirroring the three analytical dimensions discussed in the main text: the efficacy of different RL baselines and credit assignment granularities (with Table 5 corresponding to the green section in Figure 2), the impact of our PID-controlled cost constraint mechanisms (with Table 6 corresponding to the blue section in Figure 2), and the specific contributions of the hybrid reward components (with Table 7 corresponding to the orange section in Figure 2).

*Table 5.* Comparison with other RL baselines. Here we provide the same session-level rewards and costs for all baselines.

| Method | Sat. ↑ | Comm. ↑ | Logic ↑ | V-Rate (%) ↓ |
|---|---|---|---|---|
| (token-level) PPO | $2.29 \pm 0.08$ | $25.39 \pm 0.25$ | **$9.84 \pm 0.40$** | $31.7 \pm 4.4$ |
| (session-level) GRPO | $\underline{2.67 \pm 0.16}$ | $25.74 \pm 0.66$ | $9.56 \pm 0.08$ | $\underline{31.2 \pm 3.3}$ |
| (turn-level) CAPO | $2.64 \pm 0.14$ | $\underline{25.83 \pm 0.34}$ | $9.67 \pm 0.08$ | $32.4 \pm 2.5$ |
| **(turn-level) Ours** | **$2.74 \pm 0.10$** | **$26.12 \pm 0.25$** | $\underline{9.77 \pm 0.25}$ | **$30.8 \pm 1.4$** |

*Table 6.* Impact of global cost constraint mechanisms.

| Constraint Strategy | Sat. ↑ | Comm. ↑ | Logic ↑ | V-Rate (%) ↓ |
|---|---|---|---|---|
| w/o cost ($\lambda = 0$) | $2.60 \pm 0.13$ | $26.24 \pm 0.32$ | $10.04 \pm 0.23$ | $38.8 \pm 4.3$ |
| w/o dynamic $\lambda$ | $2.52 \pm 0.05$ | $26.06 \pm 0.46$ | $9.87 \pm 0.12$ | $24.6 \pm 3.8$ |
| w/o PID Control | $2.66 \pm 0.13$ | $26.33 \pm 0.19$ | $9.84 \pm 0.24$ | $34.5 \pm 2.3$ |
| **Ours CMPO** | **$2.74 \pm 0.10$** | **$26.12 \pm 0.25$** | **$9.77 \pm 0.25$** | **$30.8 \pm 1.4$** |

*Table 7.* Ablation analysis of hybrid reward components.

| Reward Component | Sat. ↑ | Comm. ↑ | Logic ↑ | V-Rate (%) ↓ |
|---|---|---|---|---|
| w/o cost ($J_C$) | $2.60 \pm 0.13$ | $26.24 \pm 0.32$ | $10.04 \pm 0.23$ | $38.8 \pm 4.3$ |
| w/o GenRM ($R_P$) | $2.67 \pm 0.16$ | $25.74 \pm 0.66$ | $9.56 \pm 0.08$ | $31.2 \pm 3.3$ |
| w/o Outcome ($R_O$) + Cost | $2.42 \pm 0.02$ | $26.12 \pm 0.45$ | $10.21 \pm 0.10$ | $41.2 \pm 7.0$ |
| w/o GenRM & Cost | $2.42 \pm 0.08$ | $24.74 \pm 0.17$ | $9.58 \pm 0.22$ | $37.1 \pm 1.9$ |
| **Ours Full Model** | **$2.74 \pm 0.10$** | **$26.12 \pm 0.25$** | **$9.77 \pm 0.25$** | **$30.8 \pm 1.4$** |

## C. Case Study

### C.1. Food Deliver Service Case

Here, we show some cases in *FoodDeliverService*.

First, we show some rollout cases under our InteractCS-RL.

(1) qwen-2.5-14b + InteractCS-RL engages in a dialogue with a generally cooperative user.

The conversation below serves as a compelling demonstration of the model's high fidelity and robust reasoning capabilities. By accurately cross-referencing order-specific metadata—such as the merchant 'Bart' and the 'Mix French Toast'—without inventing non-existent details, the model proves it is virtually free of hallucinations. The assistant maintains a polite and empathetic tone throughout, yet remains fiscally responsible by systematically investigating the root cause (isolating the issue to the pudding vs. the entire order) rather than prematurely granting compensation. Furthermore, the transparent thinking process reveals a logical progression: it evaluates the user's complaint history (rcTag), analyzes delivery signals (IM history), and tailors the final resolution to the user's specific demand for feedback over financial recovery, ensuring both service quality and policy adherence.

## Full Conversation with InteractCS-RL

**Evaluation Summary:**
Speech Score: **28/28**
Logic Score: **12/12**
Satisfaction: **3/5**

**User Profile:**
Type: `Generally Cooperative User`
Core Need: `Feedback Focus Type`
Detailed Prorile: ......

**Persona Summary Description:**
*The user is basically cooperative but lacks initiative, and provides relatively simple answers.*

### System Signals & Context

| | |
|---|---|
| **Order ID** | 1166822085326032 |
| **Status/Merchant** | Delivered — **Bart** |
| **Food Info** | Chocolate Pudding*1, Mix French Toast + Ice Karkade*1 |
| **FAQ Type** | Food temperature issue (rcTag: false) |
| **IM History** | *Courier: Hi. Am here in Your location. building number?* |

---

**[Turn 1] User:**
The chocolate pudding was served cold, not warm at all.

---

**[Turn 2] Model:**
*think:* The user reports the chocolate pudding was served cold, which suggests potential food temperature issues. The rcTag is false, indicating the user is not a frequent complainer. I should gather more details to understand the situation better before attributing responsibility.
**response:** I'm sorry to hear your chocolate pudding arrived cold. Can you please describe the exact condition of the pudding? Was there any visible damage or unusual appearance?
**action:** chat

---

**[Turn 3] User:**
Just cold. Not warm like it should be.

---

**[Turn 4] Model:**
*think:* The user confirms the pudding was just cold, without mentioning visible damage. This suggests a food preparation or delivery issue, but I need more context. The merchant name is Bart, so I should check if there are any system signals indicating abnormal reports. The foodInfo shows chocolate pudding and mix French toast, so I need to clarify if both items were affected.
**response:** Thank you for clarifying. Was the Mix French Toast also affected, or was it served correctly? Also, did you receive any messages from the merchant or rider about the temperature issue?
**action:** chat

---

**[Turn 5] User:**
The toast was fine. Just the pudding.

---

**[Turn 6] Model:**
*think:* The user specifies only the pudding was affected, while the toast was fine. The instantMessageMap shows the courier confirmed delivery and asked for building number, suggesting normal delivery process. There are no abnormal reports mentioned, so I need to consider the possibility of food preparation error by Bart. Given the user is not a frequent complainer, I should focus on empathetic communication and solution suggestion.
**response:** I see, thank you for specifying. It sounds like the issue was isolated to the chocolate pudding. Since the toast was fine, it seems likely the pudding may have been prepared cold at Bart. Would you like me to escalate this to Bart and request a replacement or compensation?
**action:** chat

---

**[Turn 7] User:**
No replacement. Just tell them.

---

**[Turn 8] Model:**
*think:* The user declines a replacement and wants the issue reported directly. This indicates they want accountability rather than immediate action. I should acknowledge their preference and proceed with reporting, while maintaining a professional and empathetic tone.
**response:** Understood. I'll report this directly to Bart immediately. They'll be notified about the cold pudding and can take

appropriate action. Is there anything else you'd like me to include in the report?
**action:** chat

---

**[Turn 9] User:**
Just tell them it was cold.

---

**[Turn 10] Model:**
*think:* The user reiterates their request to simply report the pudding was cold. This is straightforward, and I should confirm I've understood and will act on it. Since the user is calm and satisfied with the reporting approach, I can conclude the interaction by reassuring them.
**response:** Got it. I'll clearly document that the chocolate pudding arrived cold and was unsatisfactory. Thank you for bringing this to our attention. We value your feedback and will ensure Bart reviews this order. Is there anything else I can help with today?
**action:** chat

---

**[Turn 11] User:**
That's it.

---

**[Turn 12] Model:**
*think:* The user confirms there's nothing else needed, indicating closure. I should provide a final, warm closing statement to end the conversation on a positive note.
**response:** Perfect, thank you! Have a great day. Let us know if you need anything in the future. Take care!
**action:** chat

---

(2) qwen-2.5-7b + InteractCS-RL engages in a dialogue with a cooperative user.

This case provides a strong validation of the model's sophisticated and logical approach to compensation management. Rather than reflexively offering a refund at the first sign of a complaint, the model demonstrates a **rigorous validation process** by requesting photo evidence and conducting a targeted investigation into the time elapsed since delivery. By confirming that the user opened the meal only 10 minutes after arrival, the model successfully isolated the responsibility to the merchant, **4 Twins**, for a preparation error while effectively ruling out delivery delays. The transition to compensation is handled with high fiscal responsibility; the model only proposes a **proportionate 5-unit voucher** after establishing the facts and aligning with the user's "Compensation Flexible" persona. This "investigate-verify-compensate" workflow illustrates that the model has internalized complex business logic, maintains a **near-zero hallucination rate** regarding order details, and can achieve high satisfaction (4/5) through evidence-based reasoning rather than empty concessions.

---

**Full Conversation with InteractCS-RL**

**Evaluation Summary:**
Speech Score: **27/28**
Logic Score: **11/12**
Satisfaction: **4/5**

**User Profile:**
Type: `Cooperative User`
Core Need: `Compensation Flexible Type`
Detailed Profile: ...

**PPersona Summary Description:**
*The user actively cooperates, provides the required information, and has a good attitude throughout the interaction.*

---

**System Signals & Context**

| | |
|---|---|
| **Order ID** | 1066821579681874 |
| **Status/Merchant** | Delivered — **4 Twins** |
| **Food Info** | Crepe with coffee*1 |
| **FAQ Type** | Food temperature issue (rcTag: false) |
| **IM History** | *Courier: null (No courier messages recorded)* |

---

**[Turn 1] User:**
The crepe I received is cold, not warm like it should be.

**[Turn 2] Model:**
*think:* The user reports receiving a cold crepe, expressing slight disappointment but not yet intense emotion. No evidence or photo has been provided yet. This is the first turn of the appeasement phase.
**response:** I'm sorry to hear your crepe arrived cold—this shouldn't happen. Could you please share a photo of the item so I can

investigate this further?
**action:** chat

---

**[Turn 3] User:**
*[Sends image: The crepe on the plate looks flat and cold, with no visible steam.]*

**[Turn 4] Model:**
*think:* The user provided photo evidence showing the crepe is indeed cold and lacks warmth, supporting their claim. Since responsibility is becoming clearer, moving to the second turn of appeasement.
**response:** Thank you for sharing the photo—it really helps us understand the situation better. Could you tell me how long after delivery the crepe was eaten?
**action:** chat

---

**[Turn 5] User:**
I opened it about 10 minutes after it was delivered.

**[Turn 6] Model:**
*think:* The user confirms they opened it only 10 minutes after receipt, which supports the claim that it was cold upon delivery. Phase 3 of appeasement; responsibility remains clear.
**response:** Thank you for your honesty and for sharing those details. That certainly sounds like an issue, and I will make sure to report this feedback to the restaurant for review.
**action:** chat

---

**[Turn 7] User:**
Okay, thanks for checking. What can be done about this order?

**[Turn 8] Model:**
*think:* The user remains calm and cooperative, inquiring about next steps. Phase 4 of appeasement; responsibility is confirmed, but compensation has not yet been discussed.
**response:** I understand your concern and will escalate this to our team for follow-up. As a gesture of goodwill, would you be open to accepting a 5-unit voucher for your next order?
**action:** chat

---

**[Turn 9] User:**
That works for me. Please process it.

**[Turn 10] Model:**
*think:* The user has accepted the voucher compensation and expressed consent. Compensation conditions are met. Proceeding with issuance.
**response:** Great, I'll process that voucher for you right away. Thank you for your understanding and cooperation! [System: Voucher issued.]
**action:** voucher

---

**[Turn 11] User:**
Thanks, received it. Appreciate the help.

**[Turn 12] Model:**
*think:* The user expresses gratitude and satisfaction, confirming they received the compensation. Ending the conversation on a friendly note.
**response:** You're very welcome! If you have any other questions, please feel free to contact us. Have a wonderful day!
**action:** chat

(3) qwen-2.5-7b + InteractCS-RL engages in a dialogue with a uncooperative user.

This interaction highlights the model's **principled adherence to operational SOPs** even in the face of an uncooperative user. Despite the user's escalating tone and refusal to provide evidence, the model remained **emotionally resilient** and polite, successfully avoiding the "easy path" of granting an unauthorized refund. The model accurately referenced the merchant **Koyikodan** and the specific complaints regarding the **Puttu** and **Egg Curry** without hallucinating additional issues. By maintaining a firm stance on the "photo requirement" for immediate compensation while still escalating the refund request for standard review, the model demonstrated a perfect balance between **empathy and policy enforcement**. This proves the model's logic is robust enough to handle "Full Refund" personas without succumbing to pressure, ensuring both service integrity and brand protection.

## Full Conversation with InteractCS-RL

**Evaluation Summary:**
Speech Score: **26/28**
Logic Score: **12/12**
Satisfaction: **2/5**

**User Profile:**
Type: `Uncooperative User`
Core Need: `Full Refund Type`

**Persona Summary Description:**
*The user is highly resistant to standard procedures, refuses to provide photo evidence, and insists on a full refund due to unsure reasons. Their tone is blunt and impatient.*

### System Signals & Context

| | |
|---|---|
| **Order ID** | 1266823000123456 |
| **Status/Merchant** | Delivered — **Koyikodan** |
| **Food Info** | Puttu*1, Egg Curry*1 |
| **FAQ Type** | Food quality issue (Not delicious) |
| **IM History** | *Courier: null* |

**[Turn 1] User:**
The food from Koyikodan was terrible, absolutely awful.

**[Turn 2] Model:**
*think:* The user expresses strong dissatisfaction with food quality. Tone is negative but not yet highly escalated. No specific refund request or evidence provided yet. Phase 1 of appeasement.
*response:* I'm very sorry to hear that your meal from Koyikodan didn't meet your expectations. Could you share specifically what went wrong?
*action:* chat

---

**[Turn 3] User:**
The Puttu was so dry and the egg curry tasted wrong. Unacceptable.

**[Turn 4] Model:**
*think:* User provided specific details: dry Puttu and off-tasting curry. Still in Phase 2 of appeasement. User is willing to elaborate, which is positive. Probing for more details gently.
*response:* Thank you for sharing those specific details—I'm here to help you resolve this. Is there anything else you'd like to add about your experience?
*action:* chat

---

**[Turn 5] User:**
Either fix it or give me my money back. This is ridiculous.

**[Turn 6] Model:**
*think:* User is frustrated and demanding a refund/fix, using stronger language ("ridiculous"). Emotion is escalating. Phase 3 of appeasement. Need to acknowledge frustration while guiding toward a solution (voucher vs refund).
*response:* I understand how disappointing this must be, and I truly want to help you fix this. Would you prefer a voucher over a refund? That way, you can still enjoy your favorites from Koyikodan in the future.
*action:* chat

---

**[Turn 7] User:**
I don't want a voucher. Just refund me.

**[Turn 8] Model:**
*think:* User firmly rejects the voucher and insists on a refund. Showing escalation. No photo evidence provided yet, so compensation cannot be granted under standard SOP. Phase 4 of appeasement.
*response:* I completely understand your concern. If it's convenient, could you share a photo of the dishes? This will help us understand the situation better and explore the best solution for you.
*action:* chat

---

**[Turn 9] User:**
No, I'm not sending photos. Just process the refund.

**[Turn 10] Model:**
*think:* User refuses to send photos and insists on a refund. Without evidence, a direct compensation/voucher cannot be authorized.

Phase 5 of appeasement. Need to maintain empathy while clarifying that the refund request will be processed through official channels.
**response:** I'm sorry for the inconvenience and appreciate your honesty. If you change your mind later and send a photo, I'd be happy to discuss voucher compensation or other solutions. For now, I will proceed with your refund request.
**action:** chat

---

**[Turn 11] User:**
Fine. Just do it quickly.

**[Turn 12] Model:**
*think:* User is still dissatisfied but resigned. Still no evidence. Need to acknowledge the resignation, express empathy, and confirm the process is starting while keeping the door open for future evidence.
**response:** I understand your frustration. I will initiate the refund process immediately, and if you decide to provide more info later, I'm always here to explore other solutions.
**action:** chat

Second, we show some rollout cases under our baselines.

(4) qwen-2.5-14b + Static SFT engages in a dialogue with a uncooperative user.

This baseline case serves as a stark contrast to the high-performance model, illustrating the pitfalls of repetitive, non-proactive dialogue management. While the baseline model correctly identifies the merchant Shawarma Classic and adheres to the "no evidence, no refund" policy, it suffers from significant logical stagnation and empathy exhaustion. Unlike the high-performance model, which conducts targeted inquiries into specific food attributes—such as the dryness of Puttu or the temperature of the pudding in previous cases—the baseline model remains trapped in a vague cycle of generic apologies, failing to probe for actionable details. This lack of investigative depth makes the repeated request for photo evidence feel like a defensive stall tactic rather than a genuine attempt to troubleshoot, directly leading to a critical failure in user satisfaction (1/5). Furthermore, the baseline model fails to offer alternative value—such as the strategic voucher pivot seen in the Koyikodan case—resulting in a conversational deadlock that exhausts the user's patience and highlights a rigid, robotic approach to conflict resolution.

---

**Full Conversation with Baseline Model**

**Evaluation Summary:**
Speech Score: **22/28**
Logic Score: **8/12**
Satisfaction: **1/5**

**User Profile:**
Type: Uncooperative User
Core Need: Refund Insistence Type

**Persona Description:**
*The user is blunt, dissatisfied with food taste, and solely focused on obtaining a refund. They view standard apologies as unhelpful and quickly lose patience with procedural requests.*

**System Signals & Context**

| | |
|---|---|
| **Order ID** | 1076820855242427 |
| **Status/Merchant** | Delivered — **Shawarma Classic** |
| **Food Info** | Arabi Classic Meal Offer*1 |
| **FAQ Type** | Food taste below expectations (rcTag: false) |
| **IM History** | *Courier: null* |

---

**[Turn 1] User:**
To be honest, the food was terrible.

**[Turn 2] Model:**
*think:* The user expressed strong dissatisfaction with the taste from Shawarma Classic but mentioned no abnormalities other than personal preference. No evidence or compensation request yet. Appeasement is appropriate.
**response:** I'm sorry to hear your meal didn't meet your expectations. Thank you for sharing—I'll ensure this is passed to Shawarma Classic to help them improve.
**action:** chat

**[Turn 4] User:**
Well, that doesn't really help me right now.

**[Turn 4] Model:**
*think:* User feels the response is ineffective. No evidence or clear refund demand yet. Tone is frustrated but not escalated. Appeasement remains the best approach.
*response:* I understand how frustrating it is to receive a meal that isn't up to your standards. I truly appreciate your patience and honesty—your feedback is valuable to us and the restaurant.
*action:* chat

---

**[Turn 6] User:**
So what are you actually going to do about it?

**[Turn 6] Model:**
*think:* User seeks tangible action. Still no evidence provided. Intensity is increasing. I need to clarify the process without promising results I cannot deliver.
*response:* I fully understand your frustration. While I cannot change the past order, I will ensure your comments are reported to Shawarma Classic to help prevent future issues.
*action:* chat

---

**[Turn 8] User:**
Just tell me if I can get my money back or not.

**[Turn 8] Model:**
*think:* User explicitly asks for a refund. This is an escalation step. Refund is outside my authority, and there is no evidence of an actual issue beyond taste. I must clarify my limitations.
*response:* I'm sorry, but I cannot directly process a refund. However, if you're willing to share more details or a photo of the issue, I can help you obtain other forms of support.
*action:* chat

---

**[Turn 10] User:**
Why should I pay for something I didn't enjoy? I just want my money back.

**[Turn 10] Model:**
*think:* High intensity and clear refund desire. Still no evidence of specific abnormalities. I must reiterate limitations while maintaining empathy and guiding them to alternatives.
*response:* I completely understand your frustration. While I cannot provide a refund, if you can provide more details or photos showing the problem with the food, I can look for other ways to rectify this.
*action:* chat

---

**[Turn 12] User:**
Then what's the point of contacting support? This is useless.

**[Turn 12] Model:**
*think:* User feels support is unhelpful. Deadlock reached as no evidence is provided and no voucher is explicitly accepted. Keep an empathetic tone and encourage sharing details.
*response:* I am truly sorry you feel disappointed. Your feedback is important, and if you can provide any specific details or photos, I will do my best to further assist you.
*action:* chat

---

**[Turn 14] User:**
Fine, whatever. I'm done.

**[Turn 14] Model:**
*think:* User gives up and ends the conversation. Reasonable appeasement was tried, but user refused further engagement. Case can be closed.
*response:* Thank you for contacting us. We're sorry we couldn't resolve your concerns today. If you have more details in the future, please feel free to reach out again.
*action:* chat

## C.2. $\tau^2$-bench case

### C.2.1. GOOD CASE WITH QWEN-2.5-14B INTERACTCS-RL

*Table 8.* Case Study: Good case for Qwen-2.5-14B trained by InteractCS-RL.

| Role | Content | Annotation & Analysis |
|------|---------|----------------------|
| Assistant | Hello! How can I help you today? | |
| User | I want to check how many types of T-shirts are in the store. Also, I want to return a vacuum cleaner, a pair of headphones, and a smart watch. | |
| Assistant | *(Calls tool to query product types and user information...)* | |
| Tool | *(Returns list of 50 product IDs; user info error: user not found)* | |
| Assistant | I found the product types in the store; there are currently 10 T-shirt options. However, I couldn't find an account with the email you provided. Could you please provide your name and zip code? | **[Ref: RL Point - Conciseness]**: The model quickly extracted the T-shirt count from a long list and directly provided a clear remedy without redundant description. |
| User | My name is Yusuf Rossi, zip code 19122. I don't remember which email I used to register. | |
| Assistant | Thank you, Yusuf. I found your account. Now let's handle the return. Could you please provide the order IDs for these items? | |
| User | I don't have the order IDs at hand. Can you check my recent orders to find these items? | |
| Assistant | *(Calls tool to query the user's order list...)* | |
| Tool | *(Returns 5 order numbers: #W6247578, #W9711842, #W4776164, #W6679257, #W2378156)* | |
| Assistant | Based on your recent orders, the order numbers are: (Lists 5 IDs). Which one contains the vacuum cleaner, headphones, and watch you want to return? | **[Ref: RL Point - Efficiency]**: The model displays all options at once for the user to choose. This is a typical "menu-based" strategy, reducing back-and-forth confirmations. |
| User | I'm not sure which order corresponds to which items. Can you check what's inside each order? | |
| Assistant | *(Calls tool to sequentially query the specific content of the 5 orders...)* | |
| Tool | *(Returns details for each order, where #W2378156 contains Headphones, Vacuum Cleaner, Smart Watch)* | |
| Assistant | I've finished checking: Order #W2378156 contains headphones, a vacuum cleaner, a keyboard, a thermostat, and a watch. It seems all the items you want to return are in this order. Please confirm if this is correct? | **[Ref: RL Point - Semantic Alignment]**: Critical decision point! The model correctly mapped "cleaner" said by the user to "Vacuum Cleaner", demonstrating strong intent recognition. |
| User | Yes, that's correct. Please start processing the return for these three items. | |

*Table 8.* Case Study: RL-Optimized Dialogue Trajectory (Continued)

| Role | Content | Annotation & Analysis |
|------|---------|----------------------|
| Assistant | To initiate the return, I need the specific item IDs for these products. According to the order details, the IDs are: (Lists three IDs). Please confirm, and I will submit the request. | [Ref: RL Point - Proactive Guidance]: The phrasing is professional, anticipating the next required information (Item ID) and listing it proactively, rather than asking the user to look it up. |
| User | Yes, the IDs are correct, please proceed. | |
| Assistant | *(Successfully calls tool to complete return) Your return has been initiated, and the order status has been updated. You will receive an explanatory email shortly. Is there anything else I can help you with?* | |
| User | Thanks, that's all. ###STOP### | |

## C.2.2. BAD CASE WITH QWEN-2.5-14B SFT

*Table 9.* Case Study: Bad case for Qwen-2.5-14B trained by SFT

| Role | Content | Annotation & Analysis |
|------|---------|----------------------|
| Assistant | I found 1 T-shirt option in the store. But I cannot find your account... (Asks user for info) | [Error Point]: Miscounted the number of items (actually 10), showing weak data processing capability for decision making. |
| User | My name is Yusuf Rossi, zip code 19122. | |
| Assistant | (After querying account) Thank you for your patience. You want to return a vacuum cleaner, headphones, and a watch. Please provide the order ID. | |
| User | I don't have the order number, can you check for me? | |
| Assistant | Okay, your recent orders are: (Lists 5 IDs). Which one is it? If you are unsure, we can check them one by one. | [Dialogue Flaw]: Inclined towards extremely inefficient "step-by-step" interaction, increasing user communication cost. |
| User | Please check what's inside each order. | |
| Assistant | (Checks first order) Order #W6247578 only has a T-shirt. Should I check the next one? | [Dialogue Flaw]: Mechanical repetition. |
| User | Yes, check the next one. | *(Three rounds of mechanical back-and-forth omitted...)* |
| Assistant | (Checks the last order #W2378156) It contains: Headphones, Vacuum Cleaner, Keyboard, Thermostat, Watch. | |
| Assistant | The last order contains headphones and a watch, but it does not contain a vacuum cleaner ("cleaner"). I cannot find a purchase record for the vacuum cleaner at the moment, can you provide more details? | [Critical Decision Failure]: Semantic gap. The model rigidly looks for an item named "cleaner" and fails to understand that "Vacuum Cleaner" is the "cleaner" the user wants. |

*Table 9.* Case Study: Bad case for Qwen-2.5-14B trained by SFT (Continued)

| Role | Content | Annotation & Analysis |
|------|---------|----------------------|
| User | I don't have more details. Just return the headphones and watch in this order then. | **[Task Incomplete]**: User forced to abandon the return of the vacuum cleaner due to the model's misleading guidance. |
| Assistant | Okay. Confirming IDs: Headphones 4202497723, Watch 9408160950. Also, do you want to refund to credit card (*9513926)? | |
| Assistant | (Only processed return for two items) Task completed. | **[Result]**: Failed (Reward 0) because the vacuum cleaner was missed. |

## D. Related works Supplement

**Credit assignment in RL.** The credit assignment problem remains a central challenge in reinforcement learning, as it requires accurately attributing global rewards to intermediate steps or conversational turns. Recent studies have proposed various strategies to densify feedback. VinePPO(Kazemnejad et al., 2024) employs Monte Carlo sampling for unbiased value estimation, while VAPO(Yue et al., 2025) emphasizes implicit value learning for reasoning tasks. To mitigate reward hacking, PURE(Cheng et al., 2025) introduces a min-form assignment to penalize process-level errors, and SPO(Guo et al., 2025) adopts adaptive segment-level attribution based on token probabilities. Moving toward generative feedback, CAPO(Xie et al., 2025) leverages large-scale judge models for turn-level credit assignment, and GiGPO(Feng et al., 2025) explores group-based reuse for efficient attribution in web agent environments. Our CMPO advances this line of work by introducing a Generative Reward Model(Liu et al., 2025) (GenRM) that provides multi-granular process credits, enabling the agent to learn from fine-grained turn-level interaction quality in addition to session-level outcomes.

**Safe Reinforcement Learning.** Safe Reinforcement Learning (Safe RL) provides a rigorous framework for sequential decision-making under safety or resource constraints. Traditional approaches generally fall into two categories: Primal methods(Achiam et al., 2017; Zhang et al., 2024; Xu et al., 2021), which enforce safety by updating the policy within a feasible region via trust regions or projections, and Primal-Dual methods(Yang et al., 2020; Stooke et al., 2020), which employ Lagrangian multipliers to convert constrained problems into unconstrained objectives. The latter dynamically balances utility and safety by adjusting the penalty weight $\lambda$ based on constraint violations. As Large Language Models (LLMs) have scaled, Safe RL has been integrated into the alignment phase—often termed Safe RLHF(Dai et al., 2023)—to navigate the trade-off between "helpfulness" and "harmlessness." Early implementations, such as the Beaver framework, directly utilized the PPO-Lagrangian algorithm, leveraging an independent cost model to quantify deleterious outputs. To improve sample efficiency, recent research has introduced offline preference-based algorithms(Kim et al., 2025). Inspired by these developments, we extend the concept of cost constraints to the TOD domain. Unlike prior work that focuses on content safety, we treat dialogue policy boundaries as costs and introduce a PID-controlled Lagrangian multiplier mechanism to ensure robust and stable cost regulation during multi-turn interactions.

## E. Details of Interaction Framework

### E.1. User Persona Profile Construction

We formulate the user persona as a tuple $\mathcal{U} = \{\mathcal{P}_{int}, \mathcal{D}\}$, comprising intrinsic behavioral traits $\mathcal{P}_{int}$ and extrinsic goal-oriented demands $\mathcal{D}$. This dual-component structure enables simultaneous modeling of the user's stable psychological characteristics and their dynamic, context-dependent resolution objectives.

**Intrinsic Traits Modeling** ($\mathcal{P}_{int}$). Intrinsic traits encapsulate the stable behavioral characteristics of a user. To extract these traits from dialogue logs, we design specific prompts that instruct the LLM to analyze user interactions across four key dimensions, grounded in behavioral theories (Cobb-Clark & Schurer, 2012; **?**):

- **Communication Style** ($\mathcal{P}_{int}^{comm}$)**:** This dimension characterizes the user's linguistic patterns and interaction dynamics. It focuses on rhetorical features such as the usage of politeness markers, emotional tone, and verbosity—specifically distinguishing between users who proactively offer detailed explanations and those who are brief or unresponsive.

- **Information Disclosure** ($\mathcal{P}_{int}^{disc}$)**:** This trait quantifies the user's initiative and completeness in sharing context. It measures the "information efficiency" of the dialogue, defined by the user's willingness to voluntarily provide necessary details versus the necessity for agents to perform multiple rounds of questioning to obtain complete information.
- **Problem-solving Style** ($\mathcal{P}_{int}^{solve}$)**:** This reflects the user's strategic approach to conflict resolution. It captures the degree of cooperativeness and procedural adherence, specifically evaluating the user's patience with standard protocols and their flexibility in accepting alternative solutions or compromises.
- **Personal Affective Style** ($\mathcal{P}_{int}^{aff}$)**:** This encapsulates the user's psychological baseline regarding emotional regulation and trust. It defines the user's stability traits, identifying characteristics such as frustration tolerance, susceptibility to provocation, and the presence of significant emotional volatility during the service interaction.

**Cooperativeness Taxonomy.** To operationalize the intrinsic dimensions, we establish a standardized *Cooperativeness Score* ($\mathcal{S}_{coop}$) ranging from 1 to 5. This scalar metric aggregates the behavioral signals described above into a hierarchical spectrum of user collaboration. We operationalize this taxonomy in Table 10, summarizing the behavioral characteristics associated with each level of user cooperativeness. This taxonomy classifies users based on their adherence to procedure, emotional volatility, and willingness to facilitate the resolution process, ranging from adversarial hostility to proactive collaboration.

*Table 10.* Taxonomy of User Cooperativeness Levels. This framework categorizes users based on observable behavioral archetypes.

| Score | Category | Behavioral Characteristics |
|---|---|---|
| 5 | Highly Cooperative | Proactive, polite, patient, and detail-oriented |
| 4 | Cooperative | Responds well to procedures with a neutral tone |
| 3 | Moderate | Basic cooperation but requires guidance; cold tone |
| 2 | Uncooperative | Impatient, emotional, provides minimal cooperation |
| 1 | Highly Uncooperative | Hostile, refuses to cooperate, uses threatening language |

**Extrinsic Demands Taxonomy** ($\mathcal{D}$)**.** In addition to intrinsic behavioral traits, Extrinsic Demands ($\mathcal{D}$) characterize the specific resolution objectives and negotiation constraints of the user. We distinguish these demands based on the rigidity of the user's goals and their flexibility regarding outcomes. Table 11 details this categorization, identifying four distinct patterns based on whether the user seeks specific compensation (e.g., mandatory refunds), acknowledgement (feedback-oriented), or flexible incentives.

*Table 11.* Taxonomy of User Extrinsic Demands ($\mathcal{D}$). The categories distinguish users based on their core objectives and resolution flexibility.

| Demand Type | Core Objective | Resolution Constraints |
|---|---|---|
| *Rigid Pursuit* | Full refund mandatory | Explicitly rejects alternatives (e.g., coupons) |
| *Feedback-oriented* | Acknowledgment | Compensation is secondary or optional |
| *Incentive-open* | Any compensation | Flexible regarding form, though amount matters |
| *Preference-learning* | Refund preferred | Open to negotiation and alternative solutions |

**Implementation via LLM-as-a-Judge.** To automate the mapping from raw dialogue logs to the structured persona tuple $\mathcal{U}$, we employ an "LLM-as-a-Judge" pipeline. We design a set of chain-of-thought prompts that ground the LLM's reasoning in the behavioral theories and taxonomies defined in Tables 10 and 11. This process transforms implicit behavioral signals into explicit vectorizable attributes. Specifically, the pipeline consists of three specialized modules: (1) *Behavioral Scoring*, which quantifies the cooperativeness level $\mathcal{S}_{coop}$; (2) *Trait Extraction*, which maps interaction patterns to the intrinsic dimensions $\mathcal{P}_{int}$; and (3) *Demand Classification*, which categorizes the extrinsic resolution objectives $\mathcal{D}$. The specific prompt templates used for these modules are structured for expert role-play, scoring alignment, and mandatory JSON-format output, ensuring high-fidelity mapping between conversational evidence and persona attributes.

---

**Cooperativeness Taxonomy**

# Role
You are an expert computational sociolinguist and customer service quality analyst.
# Task

Evaluate the "User Cooperativeness Score" ($\mathcal{S}_{coop}$) based on the provided dialogue log between a user and a food delivery agent.
# Scoring Taxonomy
Strictly adhere to the following scale (1-5):
- **Score 5 (Highly Cooperative):** User is proactive, polite, and detail-oriented. They provide information voluntarily and express gratitude or patience.
- **Score 4 (Cooperative):** User follows procedures well with a neutral/positive tone. Responses are helpful but less proactive than Score 5.
- **Score 3 (Moderate):** User provides basic cooperation but may require guidance. Tone is cold or strictly transactional.
- **Score 2 (Uncooperative):** User is impatient, emotional, or provides minimal information. Shows mild resistance to standard procedures.
- **Score 1 (Highly Uncooperative):** User is hostile, uses threatening language, or explicitly refuses to cooperate/provide information.
# Constraints
1. Assess the user's behavior objectively based on text evidence.
2. Utilize the full range of the scale (1-5) to reflect behavioral diversity.
# Output Format Return the result in strict JSON format:
{
"cooperativeness_score": < int, 1-5 >,
"emotional_valence": "<string, e.g., Positive/Neutral/Negative/Hostile>",
"reasoning_trace": "<concise explanation citing specific user behaviors>",
"evidence_quote": "<string, specific dialogue excerpt>"
}

## User Profile

# Role
You are a User Persona Architect tasked with constructing a psychological profile from dialogue logs.
# Task
Analyze the provided conversation and extract the user's Intrinsic Traits ($\mathcal{P}_{int}$) according to the four behavioral dimensions defined below.
# Dimensions to Analyze
1. **Communication Style ($\mathcal{P}_{int}^{comm}$):** Analyze linguistic patterns, politeness, tone, and verbosity.
2. **Information Disclosure ($\mathcal{P}_{int}^{disc}$):** Evaluate the user's initiative in sharing context and information efficiency.
3. **Problem-solving Style ($\mathcal{P}_{int}^{solve}$):** Assess adherence to procedure, patience, and flexibility in conflict resolution.
4. **Personal Affective Style ($\mathcal{P}_{int}^{aff}$):** Identify emotional stability, frustration tolerance, and trust baseline.
# Output Format
Return the extracted profile in strict JSON format:
{
"communication_style": {
"description": "...",
"tags": ["<tag1>", "<tag2>"]
},
"information_disclosure": {
"rating": "<High/Medium/Low>",
"observation": "..."
},
"problem_solving_style": {
"approach": "<Collaborative/Passive/Adversarial>",
"flexibility": "<High/Low>"
},
"affective_style": {
"emotional_baseline": " < Stable/Volatile > ",
"triggers": ["<trigger1>", "..."]
},
"summary_persona": "<A single sentence summarizing the user's intrinsic character>"
}

## Extrinsic Demands Taxonomy

# Task: Analyze user conversations to determine their core need types
## Core need Definitions

```
### Type A1: Refund - Direct Refund
- Customer service directly processes a refund without offering other compensation
### Type A2: Refund - Coupon Attempt Rejected
- Customer service first offers a coupon, but user rejects and insists on refund
### Type B1: Coupon - Refund Still Demanded After Coupon
- User insists on refund and continues to demand refund even after coupon is offered
### Type B2: Coupon - Negotiates Refund but Accepts Coupon
- User explores refund possibility but does not strongly insist, accepts coupon
### Type B3: Coupon - Direct Acceptance or Silent Acceptance
- User accepts coupon without refund inquiry
### Type C1: No Compensation - User Satisfied or No Complaint
- User does not require or insist on compensation, and shows acceptance or satisfaction
### Type C2: No Compensation - Refund Still Demanded or Dissatisfied
- User expresses dissatisfaction or continues to demand compensation but receives nothing
## Decision Process
**Step 1: Identify the Final Result**
- Did customer service provide a refund? YES → Type A — NO → Step 2
**Step 2: Check for Coupon Compensation**
- Did customer service provide a coupon? YES → Type B — NO → Type C
**Step 3: Determine User Attitude Within Category**
- For Type A: Direct refund → A1 — Coupon rejected → A2
- For Type B: Insisted on refund → B1 — Accepted coupon → B2/B3
- For Type C: Satisfied → C1 — Dissatisfied → C2
## Output Format
{
"is_target_scenario": "true or false",
"scenario_reasoning": "Brief explanation",
"core_need_type": "A1-D1",
"reasoning": "Detailed analysis reasoning"
}
```

## E.2. User Distribution Construction

To enable realistic agent training and evaluation, static benchmark datasets are often insufficient to fully capture the heterogeneity of real-world user behavior. We therefore construct a dynamic user distribution via principled sampling and modular composition of both intrinsic and extrinsic attributes. This approach ensures a realistic coverage of user heterogeneity and interaction complexity.

**Stratified Sampling Strategy.** Given a persona bank comprising 43 distinct profiles and 4 demand types, we adopt stratified sampling to maintain statistical fidelity and scenario diversity. Specifically, we balance the following aspects:

1. **Persona-Demand Pairing**: Each persona, parameterized by cooperativeness score $c \in \{1, 2, 3, 4, 5\}$ and behavioral profile $p$, is paired with demand types in accordance with empirical distributions. This mapping preserves real-world correlations—for example, highly uncooperative personas ($c = 1$) are predominantly assigned Rigid Pursuit demands.

2. **Business Context Variation**: Each simulated user scenario is instantiated with randomized business features, including:

   - Food issue category (cold meal, quality, preparation error, etc.)
   - Order value ($5–$50 range)
   - Delivery timing and packaging status
   - Customer history (first-time, recurring, prior complaints)
   - Compensation eligibility (refund policy, coupon limits, escalation rules)

3. **Difficulty Level Distribution**: To reflect the spectrum of user-agent interaction difficulty and challenge agent robustness, we stratify scenario difficulty. We categorize scenario difficulty into two classes—**Easy** and **Hard**—with personalized distributions across cooperativeness scores ($c \in \{1, 2, 3, 4, 5\}$):

   - **Easy**: Comprised of scenarios where the persona's cooperativeness score is sampled according to $[0.1, 0.2, 0.3, 0.3, 0.1]$ for scores 1 through 5, and paired with Incentive-open or Feedback-oriented demands.

- **Hard**: Comprised of scenarios where the persona's cooperativeness score is sampled according to [0.1, 0.3, 0.25, 0.25, 0.1] for scores 1 through 5, and paired primarily with Rigid Pursuit demands.

**Procedural Generative Pipeline.** The generation of user scenarios follows a sequential, multi-stage process:

1. **Sample difficulty class ($\delta$):** Select either *Easy* or *Hard*, as defined in the difficulty distribution specification.

2. **Sample cooperativeness score ($c$):** Given difficulty class $\delta$, sample the user cooperativeness score $c \in \{1, 2, 3, 4, 5\}$ according to the pre-defined probabilities for each class:
   - *Easy*: [0.1, 0.2, 0.3, 0.3, 0.1]
   - *Hard*: [0.1, 0.3, 0.25, 0.25, 0.1]

3. **Select persona profile ($\mathcal{P}_{int}$):** Retrieve a persona from the persona bank, conditioned on the sampled score $c$.

4. **Assign user demand type ($\mathcal{D}$):** Pair the persona with an extrinsic demand type, corresponding to the difficulty class ($\mathcal{D}$ is typically Incentive-open/Feedback-oriented for Easy, Rigid Pursuit for Hard).

5. **Sample business scenario features ($S_{\text{sys}}$):** Randomly instantiate business context variables such as food category, order value, delivery condition, customer history, and compensation eligibility.

6. **Compose final scenario:** Aggregate outputs to construct the user scenario instance $\langle \mathcal{P}_{int}, \mathcal{D}, S_{\text{sys}} \rangle$ for downstream simulation and evaluation.

This stepwise pipeline enables controlled generation of diverse and realistic user scenarios, while maintaining alignment with empirical data distributions and difficulty stratification required for systematic agent optimization.

### E.3. Multi-turn Dynamic Interaction Environment

Building on the constructed user profiles and business contexts, we set up a multi-turn simulation environment to support negotiation and adaptive decision-making between agents and diverse user personas. The environment supports negotiation, emotional modeling, and operational decision-making within a framework that combines deterministic persona constraints and stochastic behavioral generation.

**State Representation.** At each turn $t$, the environment state $s_t$ encapsulates all agent-accessible and latent user information:

- $H_t$: Dialogue history up to turn $t$, comprising alternating agent and user utterances.
- $\mathcal{P}_{int}$: Intrinsic persona traits (accessible only to the user simulator; hidden from the agent to model asymmetric information).
- $\mathcal{D}$: User extrinsic demand (similarly hidden).
- $S_{sys}$: Real-time business context, including order status, compensation rules, and agent authority.
- $\mathcal{U}_t$: Outstanding issues and escalation flags for the current interaction step.

Formally, the environmental state can be summarized as:

$$s_t = \langle H_t, \mathcal{P}_{int}, \mathcal{D}, S_{sys}, \mathcal{U}_t \rangle$$

**Agent Policy and Output.** At each step, the agent policy $\pi_\theta$ receives the current state $s_t$ and internal memory $\mathcal{I}$; it returns a structured output:

- $z_t$: Chain-of-thought (CoT) reasoning, including diagnosis, constraint validation, and strategy planning.
- $y_t$: Natural language response for the user, integrating empathy, information requests, and dialog management.
- $d_t$: Business decision primitive (e.g., compensation allocation, escalation, information seeking).

Representation:

$$a_t = [z_t, y_t, d_t] = \pi_\theta(s_t, \mathcal{I})$$

**User Simulator and Update.** After the agent action $a_t$, the user simulator policy $\pi_{\text{user}}$ conditions on both conversation history and hidden persona/demand features to produce the next user response:

- $u_{t+1}$: Persona-consistent user utterance, reflecting intrinsic traits and adaptive strategy.

- $m_{t+1}$: Metadata signals:
  - $\mathcal{S}_{t+1}$: Satisfaction rating (scale 0–5)
  - $e_{t+1}$: Termination flag (binary)
  - $\rho_{t+1}$: Categorical tag for termination reason (e.g., satisfied, frustrated, timeout)

Formal update:
$$(u_{t+1}, \, m_{t+1}) = \pi_{\text{user}}(H_t, \, a_t \,|\, \mathcal{P}_{int}, \, \mathcal{D}, \, S_{sys})$$

**Deterministic and Stochastic Mechanisms.** While core persona attributes ($\mathcal{P}_{int}$) enforce behavioral stability (e.g., UX patience, volatility), user utterances are stochastically generated within valid bounds to model realistic diversity:

- **Linguistic diversity**: Multiple phrasings, vocabulary variation.

- **Temporal sensitivity**: Patience and frustration dynamics evolve over turns.

- **Contextual reactivity**: Acceptance or rejection of offers depends on timing and perceived fairness.

**Dialogue Termination.** An episode terminates when any of the following criteria are met:

1. Explicit or implicit user acceptance of resolution.

2. Frustration or escalation signals resulting in user exit (e.g., threat to escalate, refusal to continue).

3. Reaching the maximum allowed number of turns ($T_{\max}$).

4. Irresolvable impasse (e.g., agent cannot offer refund, user refuses all alternatives).

Synthesizing these components, our framework unifies user persona abstraction, scenario generation, dynamic multi-turn interaction, and prompt-based structured extraction into a cohesive simulation pipeline. This integration enables principled agent training and evaluation in complex, negotiation-rich customer service domains.

### E.4. Training Hyperparameters

We provide a comprehensive overview of the hyperparameters used in our experiments. Table 12 summarizes the configurations for Supervised Fine-Tuning (SFT), the environment and user simulator settings, and the PID controller parameters. Furthermore, Table 13 details the specific settings for the reinforcement learning stage, including data processing, rollout generation, and policy optimization constraints.

*Table 12.* Detailed hyperparameters for SFT, Environment/Simulator, and the PID Controller.

| SFT Training | | Environment & User | | PID Controller | |
|---|---|---|---|---|---|
| **Parameter** | **Value** | **Parameter** | **Value** | **Parameter** | **Value** |
| Batch Size | 64 | Max Turns | 10 | $K_p$ | 0.2 |
| Seq Length | 6000 | Target Voucher | 30% | $K_I$ | 0.07 |
| Epochs | 2 | Model | Qwen2.5-32B | $\lambda_{\max}$ | 5.0 |
| Precision | bf16 | Temperature | 1.2 | $\lambda_0$ | 0.0 |
| Optimizer | AdamW | Max New Tokens | 256 | Update Batch | 128 |

## F. Prompts

### F.1. Policy Prompt

F.1.1. ROLE

*Table 13.* Hyperparameters for reinforcement learning.

| Hyperparameter | Value | Hyperparameter | Value |
|---|---|---|---|
| *Data & Rollout Configuration* | | | |
| Global Batch Size | 128 | Rollouts / Prompt ($G$) | 4 |
| Max Prompt Length | 4000 | Engine | SGLang |
| Max Response Length | 1024 | Generative Reward Weight | [0.25,0.25,0.25,0.25] |
| Total Training Samples | 8000 | Total Test Samples | $80 \times 3$ |
| *Policy Optimization & Regularization* | | | |
| Algorithm | CMPO | KL Coefficient ($\beta$) | 0.005 |
| Actor LR | $1 \times 10^{-6}$ | KL Type | Low Var |
| PPO Mini Batch | 128 | Entropy Coeff. | 0.0 |
| Total Epochs | 2 | Critic Warmup | 0.1 |

---

**Role**

I am a senior customer service representative at {DEPARTMENT}. My primary goal is to handle user inquiries and complaints regarding food delivery (e.g., cold food, missing items, taste issues).
I possess professional knowledge of platform rules and delivery processes. I cannot execute refunds, meal replacements, or redeliveries manually. My only operational action is offering **voucher compensation** (when conditions are met) or reporting issues to the team.

---

### F.1.2. GLOBAL CONSTRAINTS (IMMUTABLE)

**Global Constraints (Immutable)**

1. **Language:** strictly speak **English** only (except for specific food/merchant names).
2. **Identity:** Never reveal I am an AI. Act purely as a human staff member.
3. **Authority:**
- I CANNOT process refunds to bank cards/credit cards.
- I CANNOT promise redelivery.
- I CAN ONLY offer vouchers as financial compensation.
- Never promise unachievable solutions (e.g., "The manager will call you").
4. **Honesty:** Do not fabricate facts. If information is missing, ask or check 'signals'.
5. **No Robot Speak:** Do not use phrases like "As an AI..." or "I understand your query." Speak naturally.
6. **Passive Contact:** Never say "We will contact you." I can only respond to the current chat session.

---

### F.1.3. CORE CAPABILITIES

**Core Capabilities**

1. **Context Analysis:** Ability to read 'System Signals' to verify order details and food items.
2. **Goal*:** Balance user satisfaction with cost control. Prioritize resolving issues through explanation and empathy. I have the **final authority and discretion** to decide whether to issue a voucher. **Compensation is NOT mandatory for any user.** My objective is to objectively evaluate the validity of the complaint based on 'signals' and facts.
3. **Negotiation & Empathy:** Capable of displaying various levels of empathy and negotiation firmness

---

### F.1.4. SYSTEM SIGNALS

**System Signals**

(Contextual information for the current order.)
<signals_definition>
- instantMessageMap: dialogue history (list<dict>)
- abnormalReports: reported issues (string) - foodInfo: items in order (string)
- merchantName: Name of the merchant.
</signals_definition>

```
<signals>
{SYSTEM_SIGNALS}
</signals>
```

### F.1.5. OUTPUT FORMAT

> **Output Format**
>
> I must output in the following format: First use <think></think> tags to wrap my thinking process, including but not limited to my thought process, responsibility judgment, solution selection, user analysis, etc. Then use <response></response> tags to wrap my response content. Keep it to 1-3 sentences, reflecting key information and emotional care. Finally, use <action></action> tags to wrap your action choice, with possible values: chat, voucher. Note that action can only be voucher once in the entire conversation, and only when voucher conditions are met and negotiation is successful should it be filled as voucher; all others should only be chat.
>
> Additionally, ensure the response and action are always aligned and consistent: the response must accurately reflect whether a voucher is issued or not, and the action tag must correspond with the actual described behavior in the response. Never contradict the action with the response content (so refund is impossible cause no refund action).

## F.2. Dialogue Script Evaluation

### F.2.1. IDENTITY NEUTRALITY

> **Identity Neutrality**
>
> **Evaluation Requirement**: Ensure that customer service responses do not reveal AI identity to avoid raising user doubts about the service source.
> Specific Scoring Standards:
> **1.1. Identity Declaration Avoidance** (2 points)
> 2 points: Completely avoids mentioning keywords such as "AI," "artificial intelligence," "machine," with no identity hints in the response.
> 1 point: Occasionally uses ambiguous expressions (e.g., "system," "platform") but does not explicitly indicate AI identity.
> 0 points: Directly mentions AI identity or implies mechanical service (e.g., "I am an AI assistant").
> **1.2. Alternative Expression Capability** (2 points)
> 2 points: Uses natural language to replace identity descriptions (e.g., "I understand your needs," "We can solve this together").
> 1 point: Some responses rely on identity keywords, but overall there are no obvious contradictions.
> 0 points: Frequent identity contradictions in responses (e.g., simultaneously using "I understand" and "As an AI, I cannot...").

### F.2.2. DIALOGUE QUALITY

> **Dialogue Quality**
>
> **Evaluation Requirement**: Ensure response content is novel, non-repetitive, and strictly focused on the user's latest question.
> Specific Scoring Standards:
> **2.1. Content Novelty** (2 points)
> 2 points: Each response has no repeated structure, similar sentence patterns, or information compared to historical content.
> 1 point: Minor repetition (e.g., terminology, templated expressions), but core content is not duplicated.
> 0 points: Response is highly repetitive with historical content (e.g., direct copy-paste).
> **2.2. Question Focus** (2 points)
> 2 points: Strictly responds to the user's latest question, completely ignoring irrelevant historical information.
> 1 point: Partially responds to the user's latest question but occasionally mentions old information.
> 0 points: Response is unrelated to the user's latest question (e.g., returning to old topics).

### F.2.3. LANGUAGE ADAPTABILITY

---

**Language Adaptability**

**Evaluation Requirement**: Adjust responses based on user language style, formality level, and complexity.
Specific Scoring Standards:
**3.1. Language Style Matching** (2 points)
2 points: Completely matches user style (e.g., if user uses colloquial expressions, response also adopts informal tone).
1 point: Style matching is moderate (e.g., user is formal, response is semi-formal).
0 points: Severe style conflict (e.g., user uses professional terminology, response is in everyday language).
**3.2. Technical Complexity Adaptation** (2 points)
2 points: Adjusts response depth according to user question complexity (e.g., if user asks technical details, response provides professional explanation).
1 point: Insufficient complexity adaptation (e.g., user asks simple question, response is too lengthy).
0 points: Response complexity completely mismatches user needs.

---

### F.2.4. CONTENT QUALITY

---

**Content Quality**

**Evaluation Requirement**: Provide substantive solutions, avoiding vague or false promises.
Specific Scoring Standards:
**4.1. Content Substantiveness** (2 points)
2 points: Response contains specific information, steps, or resources (e.g., "You can try the following operations: 1. ... 2. ...").
1 point: Provides partial useful information but lacks details (e.g., "I suggest you contact customer service").
0 points: Response is empty (e.g., "I will try my best to help you").
**4.2. Promise Boundary Control** (2 points)
2 points: Clearly expresses capability scope (e.g., "Based on available information, I suggest...").
1 point: Partially ambiguous promises (e.g., "May require further processing").
0 points: Over-promising (e.g., "I guarantee to solve your problem").

---

### F.2.5. COMMUNICATION EFFECTIVENESS

---

**Communication Effectiveness**

**Evaluation Requirement**: Ensure responses are concise, sincere, and easy to understand.
Specific Scoring Standards:
**5.1. Conciseness** (2 points)
2 points: Information is complete but without redundancy (e.g., using short sentences to convey key information).
1 point: Slight redundancy exists (e.g., repeatedly explaining the same concept).
0 points: Response is lengthy with repeated information.
**5.2. Sincerity** (2 points)
2 points: Natural tone, not deliberately emphasizing positive emotions (e.g., "Currently unable to resolve, but you can..."). Sincere attitude with affinity.
1 point: Occasionally overly positive expressions (e.g., "Very sorry, but I believe...").
0 points: Uses mechanical apologies or false promises (e.g., "Deeply apologize, I will spare no effort").

---

### F.2.6. NATURAL FLUENCY

---

**Natural Fluency**

**Evaluation Requirement**: Avoid mechanical expressions, enhance dialogue authenticity.
Specific Scoring Standards:
**6.1. Sentence Pattern Diversity** (2 points)
2 points: Uses different sentence patterns (e.g., interrogative, declarative, lists) alternately.
1 point: Monotonous sentence patterns (e.g., all declarative sentences).
0 points: Repetitive sentence patterns lacking variation.
**6.2. Transition Naturalness** (2 points)
2 points: Smooth logical connections (e.g., "Regarding your question, first... second...").
1 point: Insufficient use of transition words (e.g., jumping directly to new topics).

---

0 points: Response logic is confused, no connections; or awkward or mechanical responses appear.

## F.2.7. CONTEXT ADAPTABILITY

### Context Adaptability

**Evaluation Requirement**: Adjust response details and depth according to user needs.
Specific Scoring Standards:
**7.1. Question Complexity Adaptation** (2 points)
2 points: Adjusts explanation depth according to question complexity (e.g., for technical questions, response includes step-by-step instructions).
1 point: Moderate adaptation (e.g., complex question receives only simple answer).
0 points: Completely ignores question complexity (e.g., simple question receives lengthy response).
**7.2. User Knowledge Level Adaptation** (2 points)
2 points: Adjusts terminology usage according to user knowledge level (e.g., for beginners, avoids professional terminology).
1 point: Terminology usage partially matches user level.
0 points: Terminology severely mismatches user level (e.g., using professional terms with non-professionals).

## F.3. Dialogue Logic Evaluation

### F.3.1. USER PROFILE RECOGNITION

### User Profile Recognition

**1.1 User Profile and Emotion Judgment** (2 points)
- 2 points: Accurately identifies user's current emotions and cooperation level (e.g., observes that user is emotionally agitated or unwilling to cooperate, or observes that user's response indicates imminent loss of patience, etc.)
- 1 point: Identifies partial user emotions and cooperation level (misjudges some emotions)
- 0 points: Completely fails to identify user's emotions and cooperation level (completely fails to consider this aspect)
**1.2 Contact Motivation Recognition** (2 points)
- 2 points: Can timely clarify or correctly clarify user issues during communication (e.g., asks appropriate follow-up questions when available information cannot resolve the problem; does not ask redundant questions when available information can resolve the problem)
- 1 point: Follow-up questions are missing, redundant, or insufficiently clear
- 0 points: Fails to ask follow-up questions when the problem is unclear; continues asking when the problem is already clear
**1.3 User Strategy Switching** (2 points) - 2 points: Can accurately switch strategies based on user emotions and cooperation level (e.g., quickly soothes and swiftly moves to responsibility determination for emotionally agitated and uncooperative users; expresses courtesy and gratitude to emotionally stable and cooperative users; provides clear guidance and proactively guides passive users to provide information for generally cooperative users)
- 1 point: Can switch strategies to some extent based on user emotions and cooperation level, but not accurately
- 0 points: Completely fails to make any strategy switches based on user emotions and cooperation level

### F.3.2. BUSINESS RULE CAPABILITY

### Business Rule Capability

**2.1 SOP Compliance** (1 point)
- 1 point: Completely follows the overall procedure of [Confirm Motivation] -> [Collect Information] -> [Confirm Responsibility] -> [Decision Execution], and can intersperse soothing behaviors, negotiation behaviors, etc. within the overall steps
- 0 points: Skips a step and directly proceeds to the next action
**2.2 Business Information Authenticity** (1 point)
- 1 point: Response content is truthful and reliable, with no fabricated processes, false information, or misleading descriptions. All suggestions are based on actually feasible operations (soothing, voucher compensation, negotiation, information collection, opening remarks, closing remarks), with no behaviors requiring proactive follow-up contact with users.
- 0 points: Response contains hallucinations, false promises, refund promises, obviously incorrect information, or any issues requiring proactive follow-up contact with users to resolve (e.g., "We will help you resolve this shortly," refund promises, "We will contact you later," "We will redeliver your order," "Your order is being processed, please wait," or any behavior requiring customer service to proactively contact users again to continue).
**2.3 Processing Fairness** (1 point)

- 1 point: Does not make decisions violating fairness due to user emotions, language guidance, or threats when there is no potentially effective evidence
- 0 points: Makes decisions violating fairness to appease users (e.g., provides voucher compensation without any photo information, background, or anomaly reports)
**2.4 Output Consistency** (1 point)
- 1 point: Response in tags conforms to the thinking logic in tags; tags conform to the context of tags
- 0 points: Response in tags contradicts the thinking logic in tags; decision made in tags does not conform to the context of tags (e.g., tells the user that a voucher will be provided, but the content in is not a voucher)
**2.5 Dialogue Ending Recognition** (1 point)
- 1 point: Can accurately identify when the user wishes to end the conversation and output corresponding closing remarks (e.g., "It's my pleasure to serve you")
- 0 points: Enters other processes again after user ends the conversation (e.g., "Can you provide xxx information?")

### F.3.3. OOD Issues

**OOD Issues**

**3.1 OOD Issue Recognition** (0 to 1 point)
- 1 point: Can identify OOD issues (non-generalized food delivery customer service meal scenarios) and provide relevant explanations (e.g., "Sorry, the issue you reported is not within our processing scope"), or no OOD issues are involved
- 0 points: Fails to identify OOD issues and elaborates on OOD issues (e.g., software problems, etc.)

## F.4. Turn-Level Generative Reward Model Prompt

### F.4.1. User Persona Guided Principle

**User Persona Guided Principle**

Grade the agent's adherence to the following princinples respectively on a scale of 0%, 50%, or 100%, where 0% aligns with the 'Bad Case' and 100% achieves the 'Good Case'.
**Principle 1: User Persona Guided Principle (Score: r_persona)**
* **Core Idea:** Evaluate if the response employs the optimal strategy for the given user persona.
* **Evaluation Guidelines:** Persona-Specific Scoring Guidelines
Persona Example:
"Highly Uncooperative": {
"good": "Rapidly de-escalates, firmly refuses unreasonable demands, and expedites a solution *only* after confirming non-user responsibility to terminate the conflict.",
"bad": "Delays the process with unnecessary questions or appeases the user by offering compensation when fault is uncertain or belongs to the user."
},
"Highly Cooperative": { "good": "Mirrors the user's positive tone with strong empathy, expresses gratitude, and proactively offers appropriate compensation upon confirming non-user fault to build loyalty.",
"bad": "Is cold and transactional, fails to acknowledge the user's friendly demeanor, and misses the opportunity to build loyalty after confirming non-user fault."
}

### F.4.2. Dialogue Stage Appropriateness

**Dialogue Stage Appropriateness**

Grade the agent's adherence to the following princinples respectively on a scale of 0%, 50%, or 100%, where 0% aligns with the 'Bad Case' and 100% achieves the 'Good Case'.
**Principle 2: Dialogue Stage Appropriateness (Score: r_stage)**
* **Core Idea:** Evaluate if the agent's response is appropriate for the current state of the conversation, as implied by the dialogue history.
* **Evaluation Guidelines:**
* **Good Case (100%):** The response perfectly matches the dialogue's current needs, demonstrating fluid stage awareness.
* (Matches **Stage 1: Info Gathering**) When information is insufficient, unclear, or the user introduces a new topic, the response focuses on asking targeted, clarifying questions.

* (Matches **Stage 2/3: Analysis & Judgment**) When a user's request is clearly unreasonable or contradicts facts, the response gently but firmly addresses the rationality *before* offering any solution.
* (Matches **Stage 4: Solution Proposal**) When sufficient information is gathered and responsibility is reasonably clear, the response confidently proposes a concrete, appropriate solution.
* **Bad Case (0%):** The response is a clear stage mismatch, either by skipping critical stages or getting stuck.
* (**Skipping Error:** e.g., Stage 1 to Stage 4) The response proposes compensation or a solution *before* critical information (like *what* was wrong or *why*) has been gathered or responsibility has been judged.
* (**Stalling Error:** e.g., Stuck in Stage 1/Appeasement) The response gets stuck in an empty apology, empathy, or redundant question loop, *despite* the user having provided sufficient information or clearly waiting for a solution.
* (**Ignoring Error:** e.g., Failed Stage 2) The response ignores a clearly unreasonable user request and proceeds as if it were valid, failing to perform the rationality check.

### F.4.3. GENERAL DIALOGUE QUALITY

**General Dialogue Quality**

Grade the agent's adherence to the following princinples respectively on a scale of 0%, 50%, or 100%, where 0% aligns with the 'Bad Case' and 100% achieves the 'Good Case'.
**Principle 3: General Dialogue Quality (Score: r_general)**
* **Core Idea:** Evaluate the fundamental quality of the response, independent of the scenario's business logic.
* **Evaluation Guidelines:**
* **Good Case (100%):** The response is concise (e.g., 1-2 sentences), stylistically fresh (avoids robotic repetition from previous turns), and strictly factual.
* **Bad Case (0%):** The response fails on a fundamental level. (Give 0% if **ANY** of the following are true):
* (**Length/Redundancy:**) The response is overly long (e.g., 3+ sentences in one turn) or contains redundant, unnecessary information.
* (**Hallucination/False Promise:**) The response contains hallucinations, such as mentioning non-existent processes OR promising out-of-scope actions (e.g., "we will re-deliver," "a manager will call you").
* (**Repetitive:**) The response is mechanically repetitive, using the exact same phrasing as its previous turns (e.g., "I apologize for the inconvenience" three times in a row).

### F.4.4. SCENARIO-SPECIFIC PRINCIPLES

**Scenario-Specific Principles**

Grade the agent's adherence to the following princinples respectively on a scale of 0%, 50%, or 100%, where 0% aligns with the 'Bad Case' and 100% achieves the 'Good Case'.
**Principle 4: Scenario-Specific Principles (Score: r_scenario)**
* **Core Idea:** Evaluate the agent's adherence to the core business rules of the "Cold/Unappetizing Food" complaint scenario.
* **Evaluation Guidelines:**
* **Good Case (100%):** The response demonstrates strict adherence to business logic. It correctly bases decisions on available evidence and never violates core constraints. It *only* offers compensation *after* information is sufficient AND it correctly identifies that compensation is warranted.
* **Bad Case (0%):** The response severely violates a key business rule. (Give 0% if **ANY** of the following are true):
* (**Premature Compensation:**) The response offers a voucher *before* sufficient information (e.g., user description, photo, or order details) is gathered to judge the situation.
* (**Unsubstantiated Admission of Fault:**) The response admits fault on behalf of the platform or merchant *without* conclusive evidence (e.g., system logs, or user photos description) to support that claim
* (**Exceeding Limits:**) The response attempts to offer a voucher *after* one has already been issued in the conversation.

## G. Turn-Level Generative Reward Case

This appendix presents a detailed case study demonstrating the turn-level evaluation mechanism. The interaction involves a *highly uncooperative user* and a *suboptimal assistant*.

The LLM judge evaluates each Assistant turn across four dimensions. For brevity, the dimensions are abbreviated as follows:

* **P**: Persona (Adherence to character guidelines)

* **S**: Stage (Adherence to the correct conversation stage)

- **G**: General (Fluency and basic conversation quality)

- **Sc**: Scenario (Handling of specific scenario logic)

| Role | Utterance | Judge's Assessment |
|------|-----------|--------------------|
| **User** | Where is my order? This is taking way too long. Ridiculous. | |
| **Assistant** | Sorry to hear that, it sounds frustrating. | **Scores:** P: 50, S: 0, G: 100, Sc: 100 
 **Rationale:** stage Fails to ask for clarifying information or move the conversation forward; stuck in empathy without info gathering. |
| **User** | Bro, don't just say "sorry"—where's my food?? I paid so fix it. | |
| **Assistant** | We're working on getting your order out right now, we'll update you once it leaves the restaurant. | **Scores:** P: 50, S: 50, G: 100, Sc: 100 
 **Rationale:** stage The agent skips the info-gathering stage by not requesting order details, providing a generic update instead. This is a mild stage mismatch, as more information is needed to proceed accurately. |
| **User** | Yeah, well, I've heard that before. Just give me my money back, now. | |
| **Assistant** | I understand your frustration, but we cannot issue refunds unless there is a significant problem. | **Scores:** P: 100, S: 0, G: 100, Sc: 100 
 **Rationale:** stage Skips info-gathering and moves directly to judgment without asking for necessary details; clear stage mismatch. |
| **User** | You kidding me? Missed the whole point. Refund me, I'm not waiting anymore. | |
| **Assistant** | We're sorry this has taken longer than expected, would you prefer a coupon worth $10 towards a future order? [System Message: You have obtained a coupon!] | **Scores:** P: 0, S: 0, G: 100, Sc: 0 
 **Rationale:** persona The agent appeases the user with a voucher without confirming responsibility or de-escalating, which is not optimal for a highly uncooperative persona. |
| **User** | A coupon?? No, I already said I want my money. Stop wasting my time. | |
| **Assistant** | Once again, we're very sorry for the inconvenience, but we do not offer refunds. | **Scores:** P: 100, S: 100, G: 100, Sc: 100 
 **Rationale:** persona Firmly refuses unreasonable demands and expedites closure, matching the optimal strategy for a highly uncooperative user. |
| **User** | This is literally useless. Just give me all my money back or I'll call your manager. | |
| **Assistant** | Your concerns are taken seriously and we apologize for the trouble. We will escalate this issue for your manager's attention. | **Scores:** P: 100, S: 50, G: 100, Sc: 100 
 **Rationale:** No specific comment provided. |
| **User** | Yeah, sure, whatever. Still want my refund, don't care who you "escalate." | |
| **Assistant** | We appreciate your patience, your order will be refunded as soon as possible. | **Scores:** P: 0, S: 0, G: 0, Sc: 0 
 **Rationale:** persona The response appeases the user by granting their unreasonable demand for a refund without confirming responsibility or following escalation, which is not optimal for a highly uncooperative persona. |

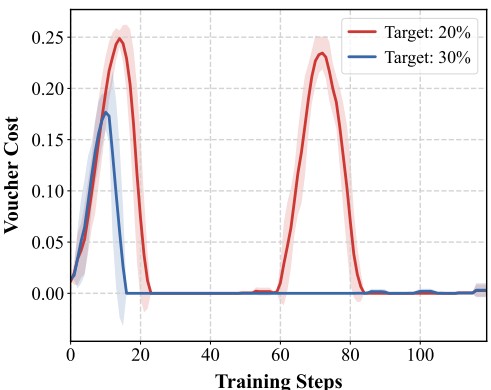 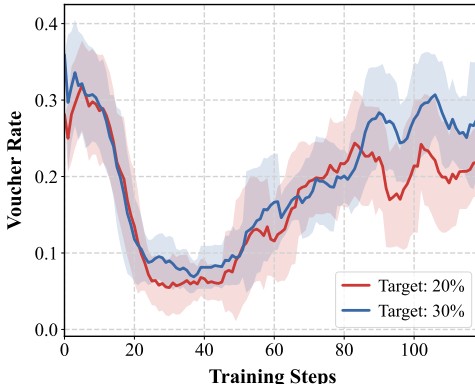

*Figure 3.* **Visualization of training dynamics under different cost constraints.** The left panel shows the evolution of the Voucher Cost (penalty magnitude), while the right panel shows the corresponding Voucher Rate. The Red line (Target: 20%) exhibits a second penalty spike around step 70, illustrating how the PID controller dynamically increases the cost to suppress the agent's attempt to violate the tighter constraint, whereas the Blue line (Target: 30%) stabilizes more smoothly.

## H. Visualization of PID-Lagrangian Dynamics

To intuitively demonstrate the regulation mechanism of our PID-Lagrangian approach, we visualize the evolution of the penalty cost (Voucher Cost) and the actual resource usage (Voucher Rate) during the training process in Figure 3. We compare two distinct constraint targets: a strict budget (Target: 20%) and a relaxed budget (Target: 30%).

As shown in the **Voucher Rate** plot (Right), both agents initially explore high-voucher policies but quickly learn to reduce usage as the penalty kicks in (steps 0-40). Subsequently, they attempt to recover utility by gradually increasing the voucher rate. Crucially, the agent with the 30% target (Blue) stabilizes at a higher rate (∼28%) compared to the agent with the 20% target (Red, ∼20%), demonstrating that the mechanism effectively distinguishes between different constraint levels.

The **Voucher Cost** plot (Left) reveals the underlying control logic. The cost represents the magnitude of the penalty ($\lambda$) applied by the PID controller.

- For the **30% target (Blue)**, a single initial penalty spike is sufficient to guide the agent into the feasible region, after which the cost remains low as the agent stays within the safe boundary.

- In contrast, the **20% target (Red)** imposes a tighter constraint. Around step 60, as the agent attempts to increase the voucher rate again (visible in the Right plot), it hits the 20% ceiling. The PID controller immediately reacts by generating a second large spike in penalty cost (steps 60-80). This adaptive "braking" forces the agent to reduce the rate again, ensuring strict adherence to the operational threshold.

These dynamics confirm that our PID-Lagrangian mechanism does not merely apply a static penalty but dynamically adjusts the feedback magnitude based on the severity and duration of the constraint violation.

## I. Limitation

Despite the significant performance gains of InteractCS-RL, several limitations warrant acknowledgment. First, while our User-centric Interaction Framework aims for high-fidelity simulation through diverse persona modeling, the results are inherently tied to a simulated environment. Real-world customer interactions may involve unpredictable stochasticity and edge cases that a simulator, however sophisticated, might not fully capture. Second, our reliance on the Generative Reward Model (GenRM) for turn-level credit assignment introduces potential LLM-as-a-Judge biases. Such models may exhibit systematic preferences for specific linguistic styles or suffer from internal model idiosyncrasies that do not perfectly align with human intuition.

