# OpenReview forum: "Reinforcing Real-world Service Agents: Balancing Utility and Cost in Task-oriented Dialogue"
_ICML.cc/2026/Conference — ICML 2026 regular_

### Official Review · Reviewer_PWMJ · 2026-03-08

**Soundness:** 2
**Presentation:** 2
**Significance:** 2
**Originality:** 2
**Overall Recommendation:** 4
**Confidence:** 4

**Summary:**

- Proposes InteractCSRL for RLHF in task-oriented dialogues, where the goals are three-fold: session-level outcome (e.g., task completion), LLM-as-a-judge evaluating turn-level conversation quality (user satisfaction, communication quality etc.), and global cost penalty (e.g., want to satisfy customers but do not want to give away too many vouchers as compensations).
- In order to study the effects of the proposed algorithm, the authors develop a new task-oriented dialogue domain based on the food delivery service.

**Compliance With Llm Reviewing Policy:**

Affirmed.

**Final Justification:**

Rebuttal responses i and ii resolved my concerns, but I agree with reviewer 6ATr that releasing the public food delivery service dataset is an important part of the paper's contribution and have adjusted my score on the condition that this dataset and benchmarks are released.

**Key Questions For Authors:**

Please review the weaknesses for detailed questions and comments. The reviewer's main concerns are around: 1. validation of metrics such as user satisfaction, communication quality, and logic quality as these are all evaluated by LLM-as-judges, 2. missing baselines for tau-2-bench and other RLHF methods for optimizing for long-horizon outcomes, and 3. clarification on Fig. 2.

**Limitations:**

Yes

**Strengths And Weaknesses:**

- Strength:
- The proposed method implemented with Qwen-2.5-7B and 14B outperform frontier models and SFT versions on the food delivery task.
- Ablations include studying the effects of different components of the proposed algorithm (such as removing the cost constraint, applying a constant penalty coefficient).
- Weakness:
- Limited baselines: PPO, GRPO, CAPO are only shown for the food delivery scenario and not evaluated on the existing tau2-benchmark (believe these should be added to Table 2). Also there are more recent RLHF methods developed specifically for multi-turn user interactions, such as Zhou et al., 2024 (https://arxiv.org/abs/2402.19446) and Wan et al., 2025 (https://arxiv.org/abs/2504.03206), Zhou et al., 2025 (https://arxiv.org/abs/2503.15478), none of which are discussed or empirically compared to. While these do not consider the global cost constraint, they still share the same goal of optimizing for a multi-turn user outcome and are specifically designed to avoid myopic agent actions, and are therefore believed to be worth discussing / comparing to.
- Suggest replacing Fig. 2 with a table with the exact numbers or include a full table in the appendix. The confidence intervals for CMPO and other baseline RL methods (GRPO, CAPO) appear to overlap on most metrics. The current bar plot makes it difficult to evaluate the actual advantages achieved by CMPO relative to the other methods, and the number of seeds / samples used to compute the error bars should be noted in the caption.
- Validation of the metrics used in Table 1 and Fig. 2 (such as user satisfaction, communication quality, and logic quality). $\pi_\text{GenPRM}$ is used both during training and for evaluation; however, since it is implemented with LLM-as-a-judge, establishing the alignment of LLM-as-judges with actual human evaluation seems critical.
- While this paper focuses on optimizing for  task-specific rewards, user-based rewards (e.g., satisfaction) and global cost constraints, the evaluation on tau-2-bench only reports task-specific metrics. It doesn't show performance on other metrics previously discussed as the primary contributions of this work. The reviewer suggests reporting performance on user satisfaction (measured by the LLM-as-a-judge, as previously done in the food delivery domain), but after validating the alignment between the LLM-as-a-judge metrics and human evaluation (please see the previous point about the lack of reliability of the LLM-as-a-judge metrics).
- In Fig 2., the average user satisfaction score achieved by CMPO w/out cost seems lower than CMPO. This seems counter-intuitive since as the authors noted "without any penalty, the agent naturally maximizes user satisfaction by overly distributing compensation" and puts the reliability of the LLM-as-a-judge for scoring user satisfaction into question.

---

> ### Author Rebuttal · Authors · 2026-03-30
>
> **Response to Reviewer PWMJ**
>
> We sincerely thank Reviewer PWMJ for the constructive review. Our detailed responses are below:
>
> **1. More Baseline Discussion**
>
> **i) Supplementary results on $\tau^2$-bench**
> We maintain our paper's setting, testing FDS-trained baselines on the $\tau^2$-bench-Retail scenario for zero-shot generalization. To address the reviewer's question on measuring FDS metrics on $\tau^2$-bench, we adopted **Comm.** and **Logic** scores. However, lacking explicit costs and rich user profiles in $\tau^2$-bench, we have to omit V-Rate and Satisfaction. The results indicate that CMPO still has certain advantages in unconstrained scenarios:
>
> | Model | Pass@1 | Comm. Rate | DB Rate | Action Reward | Comm. (from FDS) | Logic (from FDS) |
> | :--- | :--- | :--- | :--- | :--- | :--- | :--- |
> | PPO | 16.2% | 64.3% | 20.3% | 178 | 26.16 | 10.52 |
> | GRPO | 20.1% | 66.2% | 21.4% | 201 | 25.94 | 10.13 |
> | CAPO | 19.3% | 67.8% | 19.2% | 197 | 26.49 | 10.37 |
> | **InteractCS-RL (CMPO)** | **21.1%** | **67.5%** | **23.7%** | **233** | **26.74** | **10.68** |
>
> **ii) Discussion on multi-turn dialogue RL baselines**
> We deeply appreciate the reviewer mentioning several RL baselines for multi-turn dialogues. We explore them as follows:
>
> We implement the curiosity-driven baseline called CURIO (InteractCS-RL w/ CU Reward) where the LLM predicts the user's profile per turn, converting prediction errors into rewards while retaining our cost constraints. Results show that in our FDS scenario, the agent repeatedly asks for details and artificially prolongs dialogues to harvest curiosity rewards, dropping user satisfaction. This indicates predicting user profiles for exploration is ill-suited for efficiency-oriented, cost-sensitive dialogues.
>
> | Model | Sat. | Comm. | Logic | FR (%) | V-Rate (%) |
> | :--- | :--- | :--- | :--- | :--- | :--- |
> | **InteractCS-RL** | **2.74** | **26.12** | **10.07** | **100.0** | **30.8** |
> | InteractCS-RL w/ CU Reward | 2.55 | 25.43 | 9.87 | 100.0 | 32.4 |
>
> Adapting **ArCHer** and **SWEET-RL** is challenging due to scenario and computational differences. SWEET-RL requires observable "training-time information" for an accurate Critic, which is feasible in programming but subjective and unobservable in our negotiation dialogues. ArCHer uses a hierarchical off-policy architecture requiring multiple Q/V-Critics; this computational overhead is prohibitive at the 7B/14B scale.
>
> In summary, CURIO, ArCHer, and SWEET-RL represent general multi-turn RL paradigms while InteractCS-RL focuses on a **constrained balance learning paradigm**. By integrating the turn-level generative reward and batch-level costs, it achieves a mixed advantage across Turn, Session, and Batch granularities. We will add a detailed discussion of these valuable baselines to the Appendix and Related Work sections.
>
> **2. Clarification on Reward and human evaluation**
>
> Due to space limits, please refer to our detailed response to **Reviewer 6ATr (R2's Q1 & Q2)**. In brief: First, the models and rubrics used for training and testing are fundamentally different, preventing reward hacking. Second, our blind human evaluation studies demonstrate a high correlation between the judge scores and human expert ratings with Spearman's $\rho$ **0.87/0.82** for Comm. and Logic, confirming the reliability of judges.
>
> **3. Clarification on Fig 2**
>
> We agree Figure 2 is overly dense. The revised Appendix will separate ablations into clear tables with specific values and standard deviations with 3 seeds. Briefly conclusion due to limited space: CMPO demonstrates the best trade-off. For Satisfaction, PPO performs poorly (PPO 2.30 / GRPO 2.67 / CMPO 2.74). For Logic, GRPO drops (PPO 9.84 / GRPO 9.56 / CMPO 10.07). CMPO excels in both, aligning perfectly with our theme of "balance."
>
> Furthermore, the reviewer acutely pointed out a counter-intuitive detail: *CMPO w/o cost* has a lower satisfaction than *CMPO*, despite our claim that "without constraints, the agent will try to increase satisfaction by over-compensating." Through a detailed analysis of the training rollouts, we found that both phenomena actually co-exist, further illustrating the complexity of real-world customer service tasks.
>
> Specifically, user satisfaction depends not only on compensation but also heavily on language and communication strategies. We observed that when costs are removed, the model finds a "shortcut" early in training: distributing excessive voucher to quickly raise average satisfaction. Converging prematurely on this sub-optimal shortcut prevents exploring advanced communication strategies (empathy, appeasement).
>
> Our environment includes "stubborn" users whose satisfaction cannot be bought, while most users will lower demands if provided emotional value. Thus, this counter-intuitive phenomenon reinforces our core argument: utilizing a mixed reward/advantage estimation like CMPO is necessary to prevent the agent from collapsing into a lazy, unconstrained policy.

---

> > ### Author Rebuttal · Reviewer_PWMJ · 2026-04-03
> >
> > I appreciate the reviewer's ablations (in particular i and ii) and also agree with reviewer 6ATr that the release of the public dataset is a crucial aspect of this work's contribution, and have adjusted my score given the author's commitment to do so, as stated "We make a firm commitment to fully open-source the FDS benchmark, including the persona bank, environment logic, and dataset, immediately in the publication manuscript version."

---

> > > ### Author Response · Authors · 2026-04-04
> > >
> > > We are truly grateful for Reviewer PWMJ's thorough engagement during the discussion and for the generous decision to fully resolve all concerns. Your detailed and constructive feedback—on baseline coverage, metric validation, and the clarity of our ablation figures—has significantly strengthened the quality of our work, and we sincerely appreciate the time and effort invested.
> > >
> > > We are especially glad that the open-sourcing commitment resonated with you and Reviewer 6ATr. We reaffirm our commitment to releasing the FDS benchmark alongside the final publication, and we will incorporate all the experimental
> > > additions and presentation improvements discussed during the rebuttal into the revised manuscript.
> > >
> > > Thank you again for your invaluable guidance.

---

### Official Review · Reviewer_994x · 2026-03-08

**Soundness:** 3
**Presentation:** 4
**Significance:** 4
**Originality:** 3
**Overall Recommendation:** 4
**Confidence:** 4

**Summary:**

The paper proposes a new framework called InteractCS-RL that effectively tackles the problem of multiple-goal achievement issue in Task Oriented Dialogues(TOD). The authors argue that in TOD methods should manage multiple objectives like empathetic communication while at the same time provide budget aware decision-making, which is a challenging tasks for current methods. They argue current methods, focus solely on problem resolution but ignore the cost optimization perspective.

As such, they propose InteractCS-RL with following salient features:
1. User-centric Interaction Framework: To allow RL policy rollouts
2. Cost-aware Multi-turn Policy Optimization(CMPO): The RL algorithm based on GRPO with hybrid advantage estimation

**Compliance With Llm Reviewing Policy:**

Affirmed.

**Final Justification:**

My concerns have been addressed

**Key Questions For Authors:**

1. I think as mentioned, my main question would regarding GRPO vs CMPO, are there perhaps any RL algorithm based changes that I might have missed?
2. A minor second question is that what were the reasons to not use perhaps the latest GPT models or other foundational models.

**Limitations:**

yes

**Strengths And Weaknesses:**

Soundness: Partially true, I think paper is technically sound, most of the arguments provided are correct, and the experimentation is section has good baselines. My main concern is with Cost-aware Multi-turn Policy optimization, as being distinguished as a new RL algorithm, but its main contribution is reward modeling in my opinion. I've discussed this more in originality part below. Another minor concern is that the baselines used for foundation models in Table 1, are relatively older foundational models, but its somewhat acceptable depending on the timeline of experimentation and paper writing.
Presentation: Yes, the paper is well written, easy to follow, and and has a well written existing literature section.
Significance: yes, the problem is of significance value to a lot of business trying to adapt the AI for TOD problems. The formulation provides a good new direction to explore balance of objectives required for TOD problems, and problem formulation itself is a big contribution.
Originality: Partially true, I think the problem formulation is new and very valuable. However, Cost-aware Multi-turn Policy optimization as being distinguished from GRPO is I think a stretch, the underlying algorithm for both is same excluding the Advantage Estimation, which is rightfully novel. As such the comparison with different RL algorithms to show strength of CMPO is I think not well justified, as I believe same reward modeling provided by the paper might be applied to other RL algorithms to get improved results. As such the reward modeling is the main contribution, not the RL algorithm itself.

---

> ### Author Rebuttal · Authors · 2026-03-30
>
> **Response to Reviewer 994x**
>
> We sincerely thank Reviewer 994x for the constructive review. We are particularly grateful for your recognition that InteractCS-RL's concept of balancing cost and reward brings positive significance to the Task-Oriented Dialogue (TOD) domain. We have consolidated our responses to your questions and concerns below:
>
> **1. Comparison between CMPO and GRPO**
>
> We strongly agree with Reviewer 994x that CMPO inherits from GRPO, adopting the powerful idea of group sampling for advantage estimation. However, we want to emphasize that CMPO's core contribution lies in how it adapts this estimation method to multi-turn and constrained MDPs.
>
> We view CMPO as a robust RL algorithm specifically tailored for constrained task-oriented dialogues. Compared to standard GRPO, there are two primary differences:
>
> * **i) Dynamic Objective Function:** As shown in Eq. 7, our objective function is optimized under specific constraints. By employing the Lagrange multiplier method, we transform the constrained problem into a single-objective optimization process with dynamic multipliers. Furthermore, in our implementation, we efficiently integrate costs into the advantage function via a PID-controlled Lagrange multiplier.
> * **ii) Multi-grained Advantage Estimation:** Standard GRPO applies advantage estimation based on the outcome reward over an entire trajectory (Session-level). In contrast, CMPO not only utilizes this Session-level reward but also introduces a Generative Reward Model (GenRM) to apply fine-grained Turn-level rewards. Consequently, the advantages differ across turns within a complete trajectory. Moreover, the penalty term derived from constraints varies based on Batch performance. Thus, our estimation process holistically encompasses three granularities: Turn-level, Session-level, and Batch-level.
>
> These two points highlight the significant differences between CMPO and GRPO in both the optimization objective and the advantage estimation process.
>
> In addition, we appreciate the reviewer's comment on the reward formulation. However, we believe that the effectiveness of our method stems from the interplay between the reward design and the GRPO architecture. Specifically, the dynamic nature of our PID-controlled Lagrangian multiplier poses a significant challenge for PPO, as the critic would face high variance when tracking shifting reward scales (Adding a seperate cost critic is a reasonable approach but also brings more load and complexity). In contrast, GRPO naturally mitigates this through relative advantage estimation. This suggests that the success of our framework relies on this non-trivial coupling rather than reward engineering alone.
>
> Furthermore, the reviewer questioned the necessity of including CAPO and PPO as baselines. Our rationale is that PPO operates at the Token-level, GRPO at the Session-level, and CAPO at the Turn-level. Comparing them demonstrates that in real-world customer service scenarios, single-granularity RL algorithms often fail to balance all competing requirements. CMPO's mixed advantage structurally accommodates Turn-level, Session-level, and Batch-level signals simultaneously. Therefore, comparing these RL baselines not only broadens the empirical verification of performance differences but also provides clear attribution at the credit assignment level.
>
> **2. Latest Model Baseline**
>
> Regarding the latest GPT-5.x series models, we found during testing that their APIs refused to answer queries in our customer complaint domain due to built-in safety filters concerning sensitive issues like user disputes and refunds. This made it impossible to complete the evaluation for these specific models.
>
> However, we have supplemented our experiments with test results from large foundation models that are fully competitive with GPT-5.2, such as Kimi-k2.5, Gemini-3.1-Pro adn Claude-Opus-4.6. The results are as follows:
>
> | Model | Sat. | Comm. | Logic | FR (%) | V-Rate (%) |
> | :--- | :--- | :--- | :--- | :--- | :--- |
> | Kimi-k2.5 | 2.16 | 27.21 | 10.85 | 83.75 | 90.0 |
> | Gemini-3.1-Pro | 2.01 | 26.36 | 10.93 | 100.0 | 50.0 |
> | Claude-Opus-4.6 | 2.03 | 25.96 | 10.82 | 100.0 | 50.0 |
> | **InteractCS-RL-14B** | **3.05** | **27.43** | **11.35** | **100.0** | **27.5** |
>
> The results show that the latest model baseline has indeed achieved a significant advantage over GPT-4.1. For example, Kimi-K2.5 achieved an extremely high level of conversational skills, while the other two models achieved a 100% completion rate. However, these models still lag behind InteractCS RL on other metrics and still cannot effectively control V-Rate, indicating that CMPO alignment is crucial for balancing user utility and cost.

---

> > ### Author Rebuttal · Reviewer_994x · 2026-04-02
> >
> > Thank you to the authors for addressing my concerns. I still have some minor reservations about the reward modeling, but they are not substantial enough to affect my overall recommendation.

---

> > > ### Author Response · Authors · 2026-04-04
> > >
> > > We sincerely thank Reviewer 994x for the thoughtful engagement during the rebuttal. We genuinely appreciate your careful reading of our responses and the constructive advice.
> > >
> > > Regarding the remaining minor reservations on reward modeling, we fully understand and take them
> > > seriously. We will further clarify the design rationale and limitations of our reward formulation
> > > in the revised manuscript to ensure the discussion is as transparent and rigorous as possible.
> > >
> > > Thank you again for your time and for the valuable feedback that has helped improve this work.

---

### Official Review · Reviewer_6ATr · 2026-03-11

**Soundness:** 3
**Presentation:** 3
**Significance:** 3
**Originality:** 3
**Overall Recommendation:** 4
**Confidence:** 4

**Summary:**

This paper proposes an agent designed for user-centric multi-turn interaction scenarios. The authors argue that such scenarios require agents to possess not only strong domain knowledge but also refined conversational strategies to effectively serve users with different persona.

To benchmark the scenario, the paper introduces FoodDeliveryService, an environment that simulates users with different personas and goals. The authors then train an agent within this environment to optimize the task. Experimental results show that RL training improves the agent's performance in task success, dialogue quality, and cost efficiency.

**Compliance With Llm Reviewing Policy:**

Affirmed.

**Key Questions For Authors:**

1. Will the FoodDeliveryService benchmark be open-sourced?

2. In Equation 5, when computing the advantage at step t, it only considers the process reward and cost penalty at step t rather than the full return. Could the author clarify the rationale behind this design? Since the action at step t may influence future rewards and penalties, it seems that future returns should also be taken into account.

**Limitations:**

yes

**Strengths And Weaknesses:**

**Strengths**:

1. The paper introduces an interesting benchmark for user-centric conversational agents. It should be a significant contribution. FoodDeliveryService simulates users across multiple dimensions, which better reflects the complexity of real-world interactions compared to traditional only task-oriented dialogue benchmarks.

2. The paper proposes several evaluation metrics, including user satisfaction, dialogue quality, and cost efficiency. These metrics are reasonable and align well with practical requirements.

3. The paper is well written and easy to follow.

**Weaknesses**:

1. The paper does not sufficiently evaluate the quality of the simulated environment. For example, it would be helpful to understand how closely the simulation reflects real-world scenarios; how accurate the evaluators in the environment are.

2. Relevant statistics of the benchmark are not provided, such as the amount of real user data used to build the environment.

3. In the experiments, the LLM evaluator used to compute task metrics appears to be the same model used as the reward model during RL training. Since the reward model may introduce its own biases, using the same model for evaluation may lead to biased test results.

4. The proposed training method, CMPO, appears to be a relatively straightforward application of GRPO, and its novelty is limited.

---

> ### Author Rebuttal · Authors · 2026-03-30
>
> **Response to Reviewer 6ATr**
>
> We sincerely thank Reviewer 6ATr for the constructive review. We are particularly grateful for your recognition that InteractCS-RL's modeling and metric construction for real-world problems represent a meaningful advancement over traditional Task-Oriented Dialogue (TOD) settings. We have consolidated our responses to your questions and concerns below:
>
> **1. Human Evaluation on Simulator and Judge Quality**
>
> We recruited 6 domain experts from a customer service team to conduct two human evaluation studies.
>
> * **Study 1: User Simulator Fidelity.** Each expert interacted with our user simulator as a service agent across 35 dialogue sessions covering diverse user profiles (total > 200 samples). After each session, they rated the fidelity of the simulator on **Humanness**, **Decision Rationality**, and **Emotion** (5-point Likert scale). The simulator achieved average scores of **4.67/5.00**, **4.52/5.00**, and **4.72/5.00**, respectively, confirming its fidelity to real customer behavior.
> * **Study 2: Blind Trajectory Scoring.** We randomly sampled **40** test trajectories per model (SFT, GPT-4.1 w/ SOP, Qwen-2.5-14B w/ CMPO). All trajectories were anonymized and shuffled. Experts independently scored each trajectory on **Comm.** and **Logic** (5-point Likert scale) in a fully blind setting. Results show CMPO achieves the highest human scores on both dimensions (**4.78/5.00** and **4.36/5.00**), consistent with the automated evaluation (**27.43/28.00** and **11.34/12.00**).
> * **Human–LLM Correlation.** On the same trajectory set, we computed Spearman's $\rho$ at the trajectory level across all 120 samples (40 × 3 models) between human scores and LLM-as-a-judge scores (DeepSeek-V3.2, which is different from the RL reward model). The correlation reaches **0.87** for Comm. and **0.82** for Logic, validating the reliability of our automated evaluation methodology.
>
> **2. Clarification on different model usage between reward and verify**
>
> We would like to clarify two points to address your concerns regarding potential reward hacking. First, the model used to provide turn-level rewards during training is Qwen2.5-32B, while the model used for the final evaluation is DeepSeek-V3.2 (accessed via API). This architectural difference inherently prevents the policy from over-optimizing to the specific quirks of the evaluation model. Second, the rubrics guiding the LLM-as-a-judge for testing are distinctly different from the criteria used to compute the reward signal during training. We will explicitly clarify this distinction between the training reward model and the testing judge in the revised manuscript.
>
> **3. Comparison between CMPO and GRPO**
>
> We respectfully clarify that the novelty of our work lies not in proposing a new policy gradient algorithm, but in formulating TOD as a Constrained MDP and designing a multi-granularity hybrid advantage estimation that unifies Turn-level process guidance, Session-level outcome utility, and Batch-level cost constraints. For a detailed technical comparison, please refer to our response to **Reviewer 3 (R3's Q1)** for a detailed comparison.
>
> **4. More details about FoodDeliverService Bench**
>
> Because the FDS benchmark involves commercially sensitive scenarios, we are currently conducting further data desensitization and compliance checks. Once finalized, the benchmark will be fully open-sourced alongside the InteractCS-RL codebase. Furthermore, as the primary focus of this paper is on the InteractCS-RL framework itself, we kept the introduction of the FDS benchmark relatively concise. To provide more context, we extracted over 10k real user interaction logs to analyze and construct the typical user profile features used in the benchmark. We will add a comprehensive statistical analysis of this dataset to the Appendix.
>
> **5. Clarification on advantage computation**
>
> We apologize if our description of the reward computation caused any confusion. The notation in Eq. 5 was intended to indicate that the outcome reward $R_{O, i}$ yields the identical value for any turn $t \in \{1, \dots, t_{\text{end}}\}$ of the $i$-th trajectory, with the discount factor set to $\gamma = 1$ (i.e., no temporal discount). This is similar to how outcome rewards are utilized in standard GRPO. Expressing this explicitly as $R_{O, i, t} = R_{O, i}$ may help clarify our intent, and we will update the notation in the revision accordingly.

---

> > ### Author Rebuttal · Reviewer_6ATr · 2026-04-02
> >
> > Thanks for the authors' response. While some of my questions have been resolved, several concerns remain insufficiently resolved and may not easily addressed in a short rebuttal.
> >
> > My primary concern is the lack of open-sourcing of the benchmark. As noted in my original review, I consider the FoddDeliveryService benchmark to be a key contribution of this work. If the benchmark is not made publicly available, its value to the academic community is significantly reduced, and the reproducibility of the work is also compromised. Although the authors emphasize the InteractCS-RL framework as the main contribution, the proposed training method CMPO is not novel compared with GRPO. The main differences lie in reward design. Given these limitations, I am inclined to lower my score to 3.
> >
> > Additionally, my Question 2 remains unresolved. I asked why the advantage calculation (Equation 5) uses the process reward at time step $t$, $R_{P, i, t}$, rather than the return at time $t$. In the rebuttal, the authors clarified the turn-level reward $R_{O, i}$, which I understand. However, my question concerns the use of the process-level reward, which has not been addressed.

---

> > > ### Author Response · Authors · 2026-04-02
> > >
> > > Dear Reviewer 6ATr,
> > >
> > > We deeply apologize for the misunderstanding in our previous rebuttal. We misread your Question 2 regarding Equation 5 and mistakenly focused on $R_{O,i}$ instead of your actual question about the process-level reward $R_{P,i,t}$. We also completely agree with your assessment that the FoodDeliveryService (FDS) benchmark is a core contribution of this work, and its public availability is crucial for the academic community.
> > >
> > > We have taken your feedback and address your remaining concerns directly below:
> > >
> > > **(1) Answer to Question 2: Why use immediate process reward $R_{P,i,t}$ instead of return?**
> > >
> > > We fully agree with the reviewer's perspective of considering the turn-level reward in terms of return. Typically, actions influence future states, necessitating the use of a cumulative return, or a **Reward-to-Go (R2G)** formulation ($\sum_{j=t}^{T} \gamma^{j-t} R_{P,i,j}$). This paradigm is usually effective when: (i) the current action has a significant causal impact on future partial rewards, or (ii) rewards are sparse and require temporal propagation to provide learning signals. However, our turn-level reward setting does not fit these characteristics for the following reasons:
> > >
> > > * **Localized Nature of Service Norms:** Unlike strategic actions aimed at long-term goals, our process reward $R_{P,i,t}$ focuses on ensuring that *every* individual response complies with professional standards (e.g., politeness, formatting, and SOP adherence). In the context of a customer service agent, being "norm-compliant" is a highly Markovian, localized property that is largely independent of preceding or succeeding responses.
> > > * **Density of Process Signals:** R2G is often a solution for reward sparsity. In InteractCS-RL, however, process rewards are densely provided at every single turn. Since the model receives immediate and consistent feedback throughout the dialogue, the necessity of introducing a look-ahead return mechanism to "densify" the signal is significantly diminished.
> > > * **Variance and Credit Assignment:** Because long-horizon planning is already considered by the global return $R_{O,i}$, applying R2G to process rewards would introduce unnecessary variance and misattribute credit. For example, a grammatical error or a rude tone at turn $t$ must be penalized immediately; it should not be mathematically "offset" or diluted by perfectly polite responses at turn $t+2$. Thus, using the instantaneous $R_{P,i,t}$ provides a precise, low-variance gradient signal that acts as an immediate reward term.
> > > * **Logical Decoupling & Empirical Validation:** Fundamentally, our design deliberately decouples the reward logic: $R_{P,i,t}$ strictly evaluates the "action quality" of the current turn, while $R_{O,i}$ evaluates the "ultimate outcome" of the entire sequence. Introducing a cumulative return calculation would re-entangle these signals, defeating the purpose of this decoupling. This is empirically supported by PPO baseline in Fig 2. PPO relies on Generalized Advantage Estimation to model the cumulative return of step-wise rewards, yet it performs worse than CMPO/GRPO. This side evidence suggests that forcing a return-style estimation is suboptimal for our decoupled multi-granularity design.
> > >
> > > **(2) Our Commitment to Open-Sourcing the FDS Benchmark**
> > >
> > > We completely agree with you: the FDS benchmark is vital to this paper's value and reproducibility. **We make a firm commitment to fully open-source the FDS benchmark, including the persona bank, environment logic, and dataset, immediately in the publication manuscript version.** The current delay is strictly due to corpotate review and data desensitization processes, which are proceeding rapidly. While strict conference review policies prohibit us from providing external repository links during the rebuttal phase, we have proactively included comprehensive demos and subsets of the benchmark within our manuscript to ensure transparency:
> > > * **Appendix A:** Outlines the simplified scenario configurations and evaluation structures.
> > > * **Appendix B:** Provides full, unedited multi-turn dialogue trajectory cases demonstrating the interaction dynamics.
> > > * **Appendix D:** Details the complete construction process, taxonomies, and distributions of the User Personas.
> > >
> > > These appendices reflect the exact structure of the data we are preparing to release. You have our absolute assurance that the benchmark will be made fully available to the community.
> > >
> > > Thank you for holding our work to a high standard and for your invaluable feedback. We hope this clarification on the RL mechanics and our firm commitment to open-sourcing adequately address your core concerns and restore your confidence and score in our work.

---

### Official Review · Reviewer_pn6o · 2026-03-13

**Soundness:** 2
**Presentation:** 3
**Significance:** 3
**Originality:** 3
**Overall Recommendation:** 5
**Confidence:** 3

**Summary:**

This paper presents a study on developing TOD agents which balance empathy/user satisfaction with operational budgets (formalized as costs). The authors introduce a framework called InteractCS-RL, which uses a persona-grounded user simulator to create a dynamic multi-turn training environment, and algorithm called Cost-aware Multi-turn Policy Optimization which builds on GRPO by accounting for both session-level and turn-level rewards, and a penalty for cost control. The authors compare models trained on their against several LLMs on a Food Delivery Service benchmark and Tau^2-bench.

**Compliance With Llm Reviewing Policy:**

Affirmed.

**Final Justification:**

The rebuttal has addressed my main concerns.

**Key Questions For Authors:**

How were cost constraints adapted to Tau^2-bench?

Did you evaluate the user simulator? How important is its quality?

**Limitations:**

The Impact Statements section is not really filled out and the limitations are not discussed in the paper.

**Strengths And Weaknesses:**

Soundness

The experimental design is mostly solid, and the authors include both SFT and RL baselines to compare against their approach. The ablations on the reward are also greatly appreciated. However, there are concerns that this is a highly specialized approach for task-oriented dialogue whereas the SFT and RL baselines are pretty general. It would be good to compare against other dialogue-specific RL algorithms like KRLS (Yu et al. EMNLP 2023). Additionally, the claim that the model surpasses SotA models like GPT-4.1 is overclaimed. To my understanding, the paper is applying a specialized RL approach to a medium sized LLM and comparing it to a zero-shot larger LLM, so it is not too surprising that it outperforms it. It would be more appropriate to at least better tailor the prompt and/or do in-context learning.

Originality

This paper itself is quite interesting and the formulation of the problem is solid and applicable to the real world. However, the work ignores some earlier work on constrained dialogue planning (e.g. Deep Dyna-Q Peng et al. EMNLP 2018).

Significance

The paper proposes a framework which can be highly useful for training task-specific conversational agents in the real world, where out-of-the-box prompted foundation models may not be an appropriate solution. Addressing the aforementioned points on soundness would greatly help to further improve this work's significance.

Presentation

In general, the paper is well-written, well-organized, and well-framed. The figures are also helpful for understanding the paper.

---

> ### Author Rebuttal · Authors · 2026-03-30
>
> **Response to Reviewer pn6o**
>
> We sincerely thank Reviewer pn6o for the constructive review and recognizing the significance of our InteractCS-RL. We have carefully consolidated our responses to your questions and concerns below:
>
> **1. More dialogue-specific baselines and related works.**
>
> We strongly agree that comparisons with RL baselines in the TOD domain are necessary. The KRLS paradigm mentioned by the reviewer decomposes multi-turn conversations into single turns based on static answer sequences and applies offline RL. In contrast, our method aims for an end-to-end paradigm that directly optimizes for multi-turn TOD through dynamic exploration and online RL.
>
> To highlight this core difference, we designed a KRLS-style baseline tailored to our scenario. Specifically, we extracted static historical dialogues from the SFT dataset, split them as in KRLS, and required the model to generate single-turn responses based on history. We then calculated a turn-level reward emphasizing generation quality and key entity matching, without any multi-turn simulation or global cost constraints. The experimental results based on Qwen2.5-7B are as follows:
>
> | Model | Sat. | Comm. | Logic | FR (%) | V-Rate (%) |
> | :--- | :--- | :--- | :--- | :--- | :--- |
> | SFT | 2.00 | 24.15 | 9.64 | 94.2 | 41.7 |
> | KRLS | 2.46 | 25.93 | 9.92 | 100.0 | 44.5 |
> | **InteractCS-RL** | **2.74** | **26.12** | **10.07** | **100.0** | **30.8** |
>
> While KRLS outperforms SFT, it falls short of InteractCS-RL on almost all metrics, and its voucher rate severely exceeds the budget. This validates our core motivation: training in a dynamic, multi-turn environment is essential for real-world service agents to effectively balance user utility and service cost.
>
> Furthermore, we appreciate the reviewer pointing out early works like Deep Dyna-Q that address constrained dialogue planning. These methods primarily rely on predefined dialogue action policies for RL. In contrast, our approach tackles this problem in an open-ended, real-world multi-turn dialogue scenario. We will add a detailed discussion of these papers to the Related Work section in the revised manuscript.
>
> **2. Additional few-shot context with SOTA Models.**
>
> While we appreciate the reviewer’s perspective regarding the performance of RL-optimized baselines, we respectfully highlight that our 7B and 14B models achieve these results despite being nearly two orders of magnitude smaller than GPT-4.1.
>
> In response to the suggestion for few-shot context evaluation, we have extended our testing by applying two expert Standard-Operating-Procedure (SOP) strategies as few-shot prompts within the ReAct framework:
>
> | Model | Sat. | Comm. | Logic | FR (%) | V-Rate (%) |
> | :--- | :--- | :--- | :--- | :--- | :--- |
> | GPT-4.1 | 1.91 | 25.29 | 10.59 | 83.8 | 70.7 |
> | GPT-4.1-SOP1 | 1.79 | 25.04 | 10.72 | 90.6 | 20.9 |
> | GPT-4.1-SOP2 | 2.04 | 25.46 | 10.64 | 86.7 | 50.4 |
> | **InteractCS-RL-14B** | **3.05** | **27.43** | **11.34** | **100.0** | **27.5** |
>
> *SOP1* adopts a relatively conservative compensation strategy, resulting in a lower v-rate causing a drop in user satisfaction. *SOP2* relaxes the compensation restrictions, which improves satisfaction but fails to effectively control v-rate. This further demonstrates that relying solely on context engineering struggles to find the optimal trade-off; explicit RL alignment is crucial for balancing user utility and cost.
>
> **3. Clarification on $\tau^2$-bench constraints.**
>
> We clarify that $\tau^2$-bench itself does *not* contain explicit cost constraints. It is a general benchmark designed to evaluate an agent's ability to solve user problems through tool and multi-turn dialogue. We introduce $\tau^2$-bench to demonstrate InteractCS-RL's generalization in unconstrained scenarios. Beyond mere cost control, CMPO optimizes the overall policy, equipping agents with superior, transferable dialogue and reasoning strategies. As evidenced by the DB Rate and Action Reward gains in Table 2, CMPO-trained agents exhibit more precise tool-use and action execution across diverse interactive environments.
>
> We agree that exploring constraints within $\tau^2$-bench is an interesting idea. A potentially viable extension could be defining the *number of tool calls* as a cost constraint, requiring the model to solve user problems within a limited budget of actions. While determining the strictly necessary number of calls to solve a specific problem remains challenging, we believe this is a promising direction for exploration.
>
> **4. Evaluation of user simulator quality.**
>
> The quality of the user simulator is crucial for our framework, thus we supple human validation experiments. Our high human scores (4.67/5.00 on Humanness) confirm that the simulator produces sufficiently realistic interactions for effective RL training. Due to space limits, please refer to our response to **Reviewer 6ATr (R2's Q1)** for detailed discussions.

---

> > ### Author Rebuttal · Reviewer_pn6o · 2026-04-04
> >
> > Thank you for your detailed reply and helpful comments. I appreciate the additional experiments. Please be sure to add these along with details on their implementation in your revisions. I have raised my score.
> >
> > One final comment is that I would again caution against overclaiming the significance of a heavily tuned task-specific model outperforming a prompted model like GPT-4. While indeed GPT-4 is a substantially larger model, 7-14B parameters is not that few to learn a single task setting, and there is a substantial body of work already demonstrating that smaller specialized models can outperform larger, prompted generalist models.

---

> > > ### Author Response · Authors · 2026-04-04
> > >
> > > We deeply appreciate the Reviewer pn6o's continued engagement and for raising the score. We also sincerely thank the reviewer for this important clarification—it helps us communicate our contributions more precisely.
> > >
> > > The GPT-4.1 comparison was included primarily to provide a reference point for the absolute performance level of the task, as GPT-4.1 is a widely recognized frontier model that readers can intuitively calibrate against. We acknowledge that a fine-tuned 7B-14B model surpassing a prompted GPT-4.1 on a domain-specific task is indeed expected and well-established in the literature, and we will revise the paper to avoid any language that could be interpreted as overclaiming this result.
> > >
> > > Thank you again for your time and for the valuable feedback that has helped improve this work.

---

### Decision · Program_Chairs · 2026-04-30

**Decision:**

Accept (regular)

**Comment:**

This paper proposes a framework for balancing user satisfaction and operational costs in task-oriented dialogue, supported by a new benchmark (in the food delivery service domain). Reviewers liked the practical relevance of the work and the effectiveness of the proposed algorithm in managing multi-turn trade-offs. During the discussion, the authors addressed concerns regarding metric reliability by providing human-validation data and strengthened the experiments with additional specialized baselines. Their commitment to open-sourcing the benchmark was recognized as a valuable contribution to the community.

While a minor technical disagreement persists regarding the use of immediate rewards instead of cumulative returns in the advantage formulation, the concerned reviewer acknowledged the authors' justification and maintained their positive recommendation. Given the strong empirical performance and the consensus on the benchmark's utility, the paper is recommended for Acceptance.